# Low-dose IL-2 reduces IL-21$^+$ T cell frequency and induces anti-inflammatory gene expression in type 1 diabetes

Jia-Yuan Zhang [1], Fiona Hamey[1], Dominik Trzupek[1], Marius Mickunas [2], Mercede Lee[1], Leila Godfrey [1], Jennie H. M. Yang [2], Marcin L. Pekalski [1], Jane Kennet [3,4], Frank Waldron-Lynch [5], Mark L. Evans [3,4], Timothy I. M. Tree[2], Linda S. Wicker [1,6], John A. Todd [1,6] ✉ & Ricardo C. Ferreira [1,6] ✉

Despite early clinical successes, the mechanisms of action of low-dose inter-leukin-2 (LD-IL-2) immunotherapy remain only partly understood. Here we examine the effects of interval administration of low-dose recombinant IL-2 (iLD-IL-2) in type 1 diabetes using high-resolution single-cell multiomics and flow cytometry on longitudinally-collected peripheral blood samples. Our results confirm that iLD-IL-2 selectively expands thymic-derived FOXP3$^+$HE-LIOS$^+$ regulatory T cells and CD56$^{bright}$ NK cells, and show that the treatment reduces the frequency of IL-21-producing CD4$^+$ T cells and of two innate-like mucosal-associated invariant T and V$_{\gamma9}$V$_{\delta2}$ CD8$^+$ T cell subsets. The cellular changes induced by iLD-IL-2 associate with an anti-inflammatory gene expression signature, which remains detectable in all T and NK cell subsets analysed one month after treatment. These findings warrant investigations into the potential longer-term clinical benefits of iLD-IL-2 in immunotherapy.

The use of much lower doses of interleukin-2 (IL-2) than employed in cancer therapy has shown clinical efficacy in a number of inflammatory and autoimmune conditions, by stimulating a subset of CD4$^+$ T cells expressing high levels of the high-affinity trimeric IL-2 receptor, designated as regulatory T (Treg) cells[1–11]. In the autoimmune disease type 1 diabetes (T1D), the genetic association with the IL-2 pathway in mice[12] and in humans[13,14], combined with preclinical studies[15,16], have provided a strong rationale for the development of low-dose (LD)-IL-2 immunotherapy. Safety and dose determination studies in T1D have been informative and underpin future clinical trials in T1D[17–20]. We have previously conducted an adaptive single-dose observational study (DILT1D), which demonstrated the sensitivity and specificity of IL-2

signalling in vivo and determined a dosing range of 0.2−0.47 × 10$^6$ IU/m$^2$ of recombinant IL-2 (aldesleukin/proleukin) to achieve Treg increases in blood in the order of 20-50% from baseline[20]. We then performed a second study (DILfrequency)[21] to determine how often to give low-dose aldesleukin to reliably induce a steady-state Treg induction whilst maintaining the selective expansion of Tregs and not effector T (Teff) cells.

Here, we explore in depth the effects and pharmacodynamics of interval administration of low-dose recombinant IL-2 (iLD-IL-2) in DILfrequency peripheral blood mononuclear cell (PBMC) samples using a targeted single-cell multiomics approach. We perform a detailed immunophenotypic characterization of both T and NK cell

[1]JDRF/Wellcome Diabetes and Inflammation Laboratory, Wellcome Centre for Human Genetics, Nuffield Department of Medicine, NIHR Oxford Biomedical Research Centre, University of Oxford, Oxford, UK. [2]Department of Immunobiology, King's College London, School of Immunology and Microbial Sciences, London, UK. [3]Wellcome-MRC Institute of Metabolic Science, Metabolic Research Laboratories, University of Cambridge, Cambridge, UK. [4]National Institute for Health Research Cambridge Biomedical Research Centre, Addenbrooke's Biomedical Campus, Cambridge, UK. [5]Vertex Pharmaceuticals, Vertex Cell & Gene Therapies, Boston, MA, USA. [6]These authors jointly supervised this work: Linda S. Wicker, John A. Todd, Ricardo C. Ferreira. ✉e-mail: john.todd@well.ox.ac.uk; ricardo.ferreira@well.ox.ac.uk

compartments, with an emphasis on the CD4[+] CD127[low]CD25[hi] T cell and CD56[bright] (CD56[br]) NK cell subsets, which have been previously shown to be highly sensitive to LD-IL-2 treatment. Aside from the expected expansions of thymic-derived FOXP3[+]HELIOS[+] Tregs and CD56[br] NK cells in T1D patients, we show that iLD-IL-2 reduces the frequency of IL-21-producing CD4[+] T cells. Furthermore, we discover that in contrast to these transient cellular alterations that are maintained only during the treatment period, iLD-IL-2 also induces prolonged transcriptional changes with anti-inflammatory properties in both T and NK cell subsets that are present one month after cessation of treatment. Taken together, our findings indicate that iLD-IL-2 can reduce IL-21 during treatment, and induce a suite of cellular alterations that promotes the establishment of a long-lived anti-inflammatory environment.

## Results

### Interval administration of low-dose IL-2 (iLD-IL-2) induces and maintains increased frequencies of CD4[+] CD127[low]CD25[hi] T cells and CD56[br] NK cells

The DILfrequency[21] study established an optimal three-day aldesleukin dosing interval, with doses ranging from $0.20–0.47 \times 10^6$ IU/m² (Fig. 1a). To gain further insight into the mechanism of action of iLD-IL-2, we selected 18 DILfrequency participants (adults with long-standing T1D) treated with this dosing regimen (Supplementary Data 1) and characterised the cellular alterations in blood by polychromatic flow-cytometry (FACS). We observed a sustained increase in the frequency of CD4[+] CD127[low]CD25[hi] T cells (henceforth designated as the CD4[+] Treg gate) during the dosing phase, but no alterations in the frequency of the broader CD4[+] conventional T (Tconv) or CD8[+] T cell populations (Fig. 1b, c), including in a subset of CD45RA[−]CD62L[−] effector memory CD4[+] T cells (T_{EM}), known to be enriched for recently activated effector T cells (Supplementary Fig. 1a–c).

Outside the T cell compartment, iLD-IL-2 also caused sustained increased frequencies of a subset of CD56[br] NK cells (Fig. 1c). This effect was more pronounced than the increase in CD4[+] Tregs and is consistent with the relatively high affinity of the CD56[br] NK subset for IL-2, especially when compared to the bulk NK population[22,23]. We observed no alteration in the frequency of the classical CD56[dim] NK cells, which represent the majority of CD56[+] NK cells in circulation. The IL-2-induced increase in CD4[+] Tregs and CD56[br] NK cells was recapitulated when assessing absolute numbers of cells in circulation, although their relative low abundance in blood resulted in no discernible alteration in the numbers of either CD4[+] and CD8[+] T cells or CD56[+] NK cells (Fig. 1d, e). An increase in both frequency and numbers of CD56[br] (but not CD56[dim]) NK cells in blood was observed in all IL-2 dose groups (Supplementary Fig. 1d–f).

### Investigating IL-2-induced cellular alterations in blood using single-cell multiomics

We next profiled PBMCs from 13 selected participants treated with the three-day interval dosing regimen (Supplementary Data 1) using a targeted single-cell multiomics approach, which enables the parallel quantification of specific mRNA and cell-surface proteins (AbSeq) in each cell[24]. To increase the power of this approach to identify small, yet potentially important, changes elicited by iLD-IL-2, we have adopted a cell enrichment strategy to greatly increase the numbers of CD4[+] Tregs and CD56[br] NK cells profiled in this study. In addition, we designed a custom mRNA panel of 565 oligonucleotide probes, assessed in parallel with the expression of 65 cell-surface protein targets (Supplementary Data 2)[25]. From each participant, we selected three longitudinal blood samples on Day 0 (baseline), Day 27 (immediately before the last IL-2 injection) and Day 55 (four weeks after the last IL-2 dose), respectively. We then isolated five immune populations from each sample following predefined proportions: 30% CD4[+] Tregs; 25% CD4[+] Tconv and CD8[+] T cells 12% CD56[br] and 8% CD56[dim] NK cells,

which were analysed with or without in vitro stimulation (Fig. 1f). Unsupervised clustering of the 482,531 unstimulated cells passing quality control (QC) revealed a clear delineation of the sorted T and NK cell populations, identifying 15 distinct functional clusters (Fig. 1g). We observed good overlap among cells from each participant, with similar representation in the five sorted immune populations (Supplementary Fig. 2a–c). A similar result was observed for the clustering of the 323,839 stimulated cells passing QC (Supplementary Fig. 2d–f), indicating minimal batch effects that are usually prevalent in scRNA-seq data.

In the unstimulated cells, we also identified two clusters characterised by the distinct co-expression of a set of canonical cell cycle genes (see Methods), including *MKI67* (encoding for Ki-67), which corresponded to cycling T and NK cells (Supplementary Fig. 3a). Overall, we found no evidence that iLD-IL-2 increased the frequency of proliferating cells. Conversely, we found a reduction in the frequency of proliferating CD56[br] NK cells and CD4[+] Tregs at Day 27 (Supplementary Fig. 3b). Given the previously reported increased frequency of Ki-67[+] Tregs and CD56[br] NK cells following LD-IL-2 treatment, we next investigated the frequency of proliferating cells by assessing the intracellular expression of Ki-67 in patients treated with either single or multiple doses of LD-IL2. In agreement with previous observations[4,6,26], we found a marked increased frequency of Ki-67[+] cells in both CD4[+] Tregs and CD56[br] NK cells following the first dose of IL-2 (Supplementary Fig. 3c, d). In the multiple dosing cohort, we noted that this increased frequency of Ki-67[+] cells persisted up to Day 27 only in CD56[br] NK, but not in the CD4[+] Treg cells (Supplementary Fig. 3d). This discrepancy between the multiomics and FACS assessment of proliferating CD56[br] NK cells may reflect a difference in the use of protein or mRNA markers to assess cell cycle progression. Both *MKI67* (mRNA) and Ki-67 protein are synthesised and accumulate during the S and G2/M phases of the cell cycle and show very good correlation in synchronised cells[27]. However, in contrast to *MKI67*, Ki-67 protein is more long-lived, and can still be detected in G1 after cell division[28]. The levels of Ki-67 protein in G1 depend on the rate of protein accumulation during the S and G2/M phases, leading to the much higher rate of detection of Ki-67[+] cells compared to the frequency of cells in the S-G/M phase (e.g. 37% vs 1% in Fraction II Tregs)[29]. Consistently, we found that 20% of total CD4[+] Tregs are Ki-67[+] at baseline, but only ~1% were identified as proliferating by multiomics analysis. These data indicate that in contrast to Ki-67[+] protein levels, mRNA quantification allows a more specific identification of actively dividing cells (in the S-G/M phases). Therefore, an explanation for the discrepancy between the FACS and multiomics assessment of proliferating CD56[br] NK cells at Day 27, is that the increased levels of Ki-67[+] cells reflect protein accumulation in recently divided G1 cells. In contrast, the decrease in proliferating cells from the multiomics analysis reflects a reduction of actively dividing cells in blood, possibly due to an increased likelihood for cells to enter the cell cycle in tissues as IL-2 accumulates on the extracellular matrix during dosing[30–33].

### Interval administration of low-dose IL-2 selectively expands FOXP3[+]HELIOS[+] Tregs

Next, we sought to investigate whether our iLD-IL-2 regimen altered the relative composition of the cells sorted from the CD127[low]CD25[hi] Treg gate (Supplementary Fig. 4). Analysis of the expression of the canonical Treg transcription factors *FOXP3* and *IKZF2* (encoding HELIOS) allowed the stratification of CD127[low]CD25[hi] Treg clusters into three functional groups according to their origin (Supplementary Fig. 5 and "Methods"). Differential abundance analysis revealed an enrichment in the frequency of naïve FOXP3[+]HELIOS[+] Tregs and a concomitant reduction of CD25[+] FOXP3[−]HELIOS[−] Teff cells on Day 27, after the conclusion of the four-week IL-2 dosing phase (Fig. 2a, b), which was replicated in stimulated cells (Fig. 2c, d). Given the observed increase in CD127[low]CD25[hi] T cells by FACS following LD-IL-2, we next

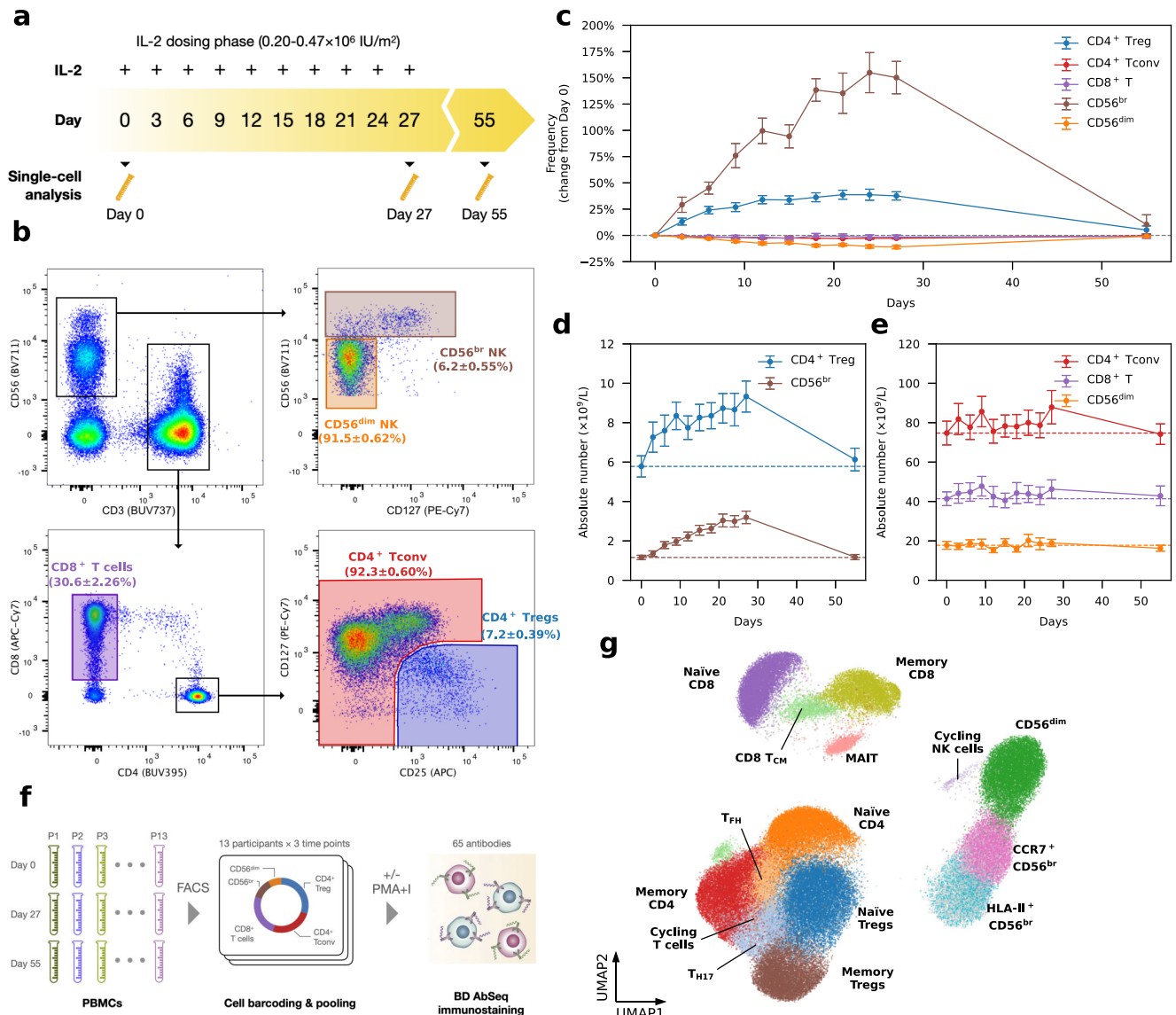

**Fig. 1 | Interval low-dose IL-2 administered every three days sustained an increased frequency of CD4+ regulatory T cells and CD56br NK cells in blood.** **a** Schematic representation of the IL-2 dosing regimen used in this study. **b** Gating strategy for the delineation of the five T and NK cell populations assessed by flow cytometry in the 18 DILfrequency study participants treated with the 3-day interval dosing interval. The frequency of each cell population at baseline (Day 0) is presented as the mean value ± SEM. **c** Variation (depicted as the % change from baseline levels) in the frequency of the five assessed immune subsets in blood (denominator is total CD3+ T cells for T cell subsets, CD4+ T cells for the CD4+ Treg subset and total CD56+ NK cells for NK subsets). Data are presented as mean values ± SEM (*n* = 18 participants). Absolute numbers of the assessed CD4+ CD127lowCD25hi (Tregs) and CD56br NK cell (**d**), as well as the broader CD4+ and CD8+ T and CD56+ NK cell (**e**) subsets in blood. Data are presented as mean values ± SEM

in *n* = 18 participants treated with either 0.2 (*n* = 4), 0.32 (*n* = 12) or 0.47 (*n* = 2) × 10⁶ IU/m² IL-2. Dotted lines indicate the average baseline numbers of each subset. Individual-level data, as shown in (**c**–**e**), are included in ref. 21. **f** Experimental design employed in this study. Equal proportions of flow-sorted cells subsets (30% CD4+ Tregs; 25% CD4+ Tconv and CD8+ T; 12% CD56br NK and 8% CD56dim NK cells) from *n* = 13 selected DILfrequency participants treated with the 3-day dosing interval were profiled by single-cell multiomics. **g** Uniform Manifold Approximation and Projection (UMAP) plot depicting the 15 clusters identified from the clustering of *n* = 482,531 unstimulated cells. Clusters were manually annotated based on the expression of key mRNA and protein markers. Light grey cells correspond to CD56br NK cells without sorting gate information. Source data for (**c**, **d**, **e**) are provided as a Source Data file.

investigated whether these changes were also reflected in the absolute numbers in circulation, which were inferred from the relative frequencies identified by single-cell analysis, scaled by the whole-blood CD127lowCD25hi Treg counts. We found that the IL-2-induced expansion on Day 27 was restricted to the thymic-derived FOXP3+HELIOS+ Treg subsets (Fig. 2e). We observed very good concordance between unstimulated and stimulated conditions, with naïve Tregs showing >2-fold increase and memory FOXP3+HELIOS+ Tregs displaying approximately a 70% increase in cell numbers compared to baseline pre-treatment levels (Fig. 2e). In contrast, despite the expansion of CD127lowCD25hi T cells, we obtained no evidence of alterations in the

number of activated CD25+ FOXP3−HELIOS− Teffs contained in this population, even though they express the high affinity trimeric IL-2 receptor (Fig. 2e). Furthermore, we observed no changes in the in vitro suppressive capacity of CD4+ CD127lowCD25hi T cells after IL-2 treatment on a per cell basis compared to the pre-treatment cells (Fig. 2f). The alterations in the relative composition of CD127lowCD25hi T cells were found to be transient and reverted to baseline one month after the last dose of IL-2 (Day 55; Supplementary Fig. 6). In contrast, we observed no notable changes in the relative abundance of CD4+CD25−/low Tconv subsets during or after IL-2 treatment (Supplementary Fig. 7).

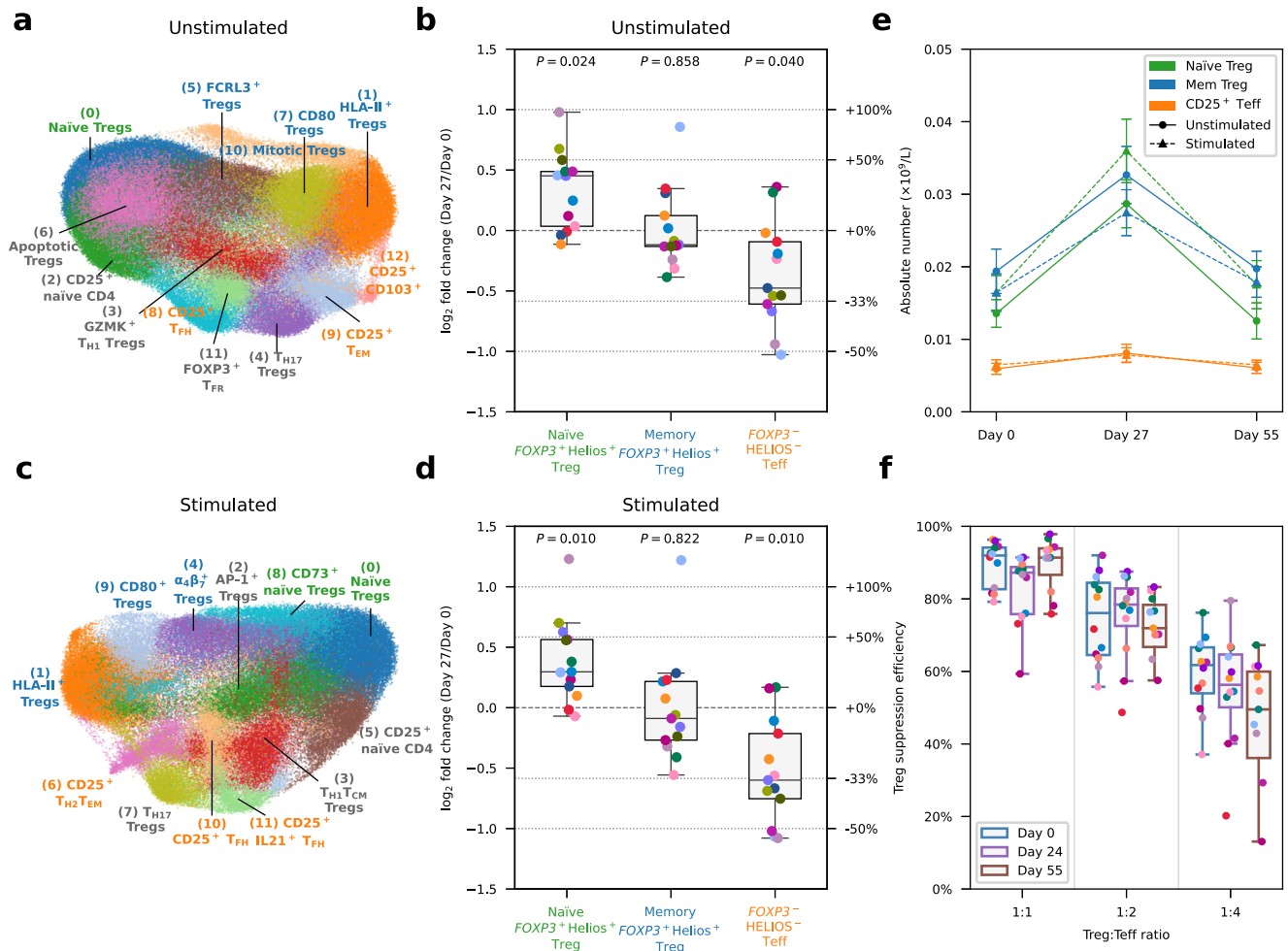

**Fig. 2 | Interval low-dose IL-2 immunotherapy selectively expands FOXP3+HELIOS+ Tregs. a** UMAP plot depicting n = 149,202 unstimulated cells sorted from the CD127lowCD25hi Treg gate. Clusters were manually annotated based on the expression of key mRNA and protein markers. Clusters corresponding to: (i) naïve Tregs (defined as CD45RA+ FOXP3+HELIOS+); (ii) memory Tregs (defined as CD45RA− FOXP3+HELIOS+); or (iii) CD25+ FOXP3−HELIOS− Teffs (defined as CD45RA− FOXP3−HELIOS−) are annotated in green, blue and orange, respectively. All other clusters not conforming to this definition are annotated in black. **b** Relative frequency changes of naïve FOXP3+HELIOS+ Treg, memory FOXP3+HE-LIOS+ Treg, and CD25+ FOXP3−HELIOS− Teff clusters on Day 27 compared to baseline (Day 0) levels, for unstimulated cells. **c, d** Corresponding UMAP plot (**c**) and relative frequency of the identified subsets (**d**) for n = 115,153 stimulated CD4+ T cells sorted from the CD127lowCD25hi Treg gate. **e** Absolute cell numbers of naïve FOXP3+HELIOS+ Tregs, memory FOXP3+HELIOS+ Tregs, and CD25+ FOXP3−HELIOS− Teffs on Day 0, Day 27 and Day 35, for unstimulated and stimulated cells. Absolute cell numbers of the Treg subsets were calculated by multiplying the absolute number of Tregs measured by FACS by the relative frequency of the respective subset measured by single-cell multiomics. Data are presented as mean values ± SEM. **f** Per-cell Treg suppression efficiency for each time point, assessed in vitro by culturing CD4+CD25hiCD127low Tregs with autologous CD4+CD25−/lowCD127+ Teffs at various ratios (x axis). n = 12 participants. In (**b**) and (**d**), P values were calculated using two-sided Wilcoxon signed-rank tests comparing the relative frequency of the subsets between Day 27 and Day 0. In (**b**, **d** and **f**), each dot represents cells from a single participant. Dots with the same colour represent the same participant. Each box ranges from the first quartile (Q1) to the third quartile (Q3), with a central line indicating the median. The bottom and top whiskers extend to the most extreme data points within Q1 − 1.5 × (Q3 − Q1) and Q3 + 1.5 × (Q3 − Q1), respectively. In (**b**, **d**, and **e**), n = 13 participants. Source data for (**b**, **d**, **e**, **f**) are provided as a Source Data file.

Using pseudo-time analysis, we identified a single trajectory of FOXP3+ Treg cell differentiation among CD4+ T cells, stemming from the conventional naïve Tregs towards the classical HLA-II+ effector Treg subset (Supplementary Fig. 8a–c). In agreement with our compositional analysis, we found a significantly increased ratio of naïve to HLA-II+ Tregs at Day 27 (Supplementary Fig. 8d). Notably, there was no alteration in the differentiation status of Tregs in response to iLD-IL-2, as illustrated by the overlapping pseudo-time distribution within the Treg differentiation trajectory between Days 0, 27 and 55 (Supplementary Fig. 8d). These data indicate that iLD-IL-2 treatment is affecting only the output and/or survival of thymic-derived naïve Tregs but not their differentiation towards HLA-II+ effector Tregs. Furthermore, we also replicated the transient IL-2-induced increase in the frequency of naïve Tregs by measuring the naïve:memory Treg ratio at Day 27 by FACS in all 18 DILfrequency participants treated with the 3-day interval dosing schedule (Supplementary Fig. 9a–c). We found good concordance between FACS and single-cell data on the frequency of naïve and memory Tregs as well as their ratios (Supplementary Fig. 9d, e). To confirm the characterisation of Treg subsets obtained through the single-cell multiomics analysis, we next assessed the protein expression of FOXP3 and HELIOS by FACS in our single and multiple dosing replication cohorts. In agreement with the multiomics analysis, we found a selective decrease in the relative proportion of CD45RA− FOXP3−HELIOS− cells within CD127lowCD25hi T cells in both cohorts in response to iLD-IL-2 (Supplementary Fig. 10a–c). These results translated into a consistent increase in the frequency of both CD4+CD25hi FOXP3+ Treg subsets and a concomitant reduction in CD4+CD25hi FOXP3−HELIOS− Teffs (Supplementary Fig. 10d, e).

In addition to the conventional CD25$^{hi}$ FOXP3$^+$ Treg subsets, we have recently shown that a subset of CD25$^{-/low}$ FOXP3$^+$ T cells are increased in patients with autoimmune diseases[34]. We found no substantial change in the frequency of CD25$^{-/low}$ FOXP3$^+$ T cells (defined as cells with > =1 copy of FOXP3 within the CD4$^+$CD25$^{-/low}$ Tconv gate; Supplementary Fig. 11a, b). We replicated these results using a FACS approach to vastly increase the number of characterised CD25$^{-/low}$ FOXP3$^+$ T cells. In agreement with the multiomics analysis, we found no change in the frequency of this rare subset in either the single or multiple dosing cohorts (Supplementary Fig. 11c–g). However, we noted a small reduction in the frequency of CD25$^{-/low}$ FOXP3$^+$ T cells at Day 27 by both methods, suggesting that iLD-IL-2 may restore the CD25 expression in these cells.

## IL-21-producing T cells are reduced by iLD-IL-2 treatment

Among the identified CD25$^+$ FOXP3$^-$HELIOS$^-$ Teff clusters within the sorted CD127$^{low}$CD25$^{hi}$ Treg cells, we noted an IL-2-induced reduction of a subset with a marked T follicular helper (T$_{FH}$) cell profile, including the expression of canonical T$_{FH}$ receptors CXCR5, PD-1 and ICOS, and the transcription factor *MAF* (CD25$^{hi}$ T$_{FH}$, cluster 8; Supplementary Figs. 4a, 6a). Although differentiated T$_{FH}$ cells are rare in blood, within cells sorted from the CD25$^{-/low}$ Tconv gate, we were able to characterise an analogous heterogeneous cluster of central memory (T$_{CM}$) CXCR5$^+$ cells (cluster 1; Supplementary Fig. 7a–c), which is known to be enriched with circulating precursors of T$_{FH}$ cells. Upon in vitro stimulation, which elicited the expression of the classical T$_{FH}$ cytokine IL-21, we obtained a better discrimination of the T$_{FH}$ clusters. This led to improved differentiation of a cluster of CXCR5$^+$ T$_{CM}$ cells (cluster 2) and a cluster of CXCR5$^{low}$ IL-21$^+$ T cells (cluster 6) that was reduced with IL-2 dosing (Supplementary Fig. 7d–f). In addition, within the CD127$^{low}$CD25$^{hi}$ Treg gate, in vitro stimulation allowed the stratification of the activated CD25$^+$ T$_{FH}$ cells into two separate clusters (cluster 10 and 11) based on their expression of IL-21, which were both reduced by iLD-IL-2 treatment (Supplementary Fig. 6b). We note that a related cluster of FOXP3$^+$ T follicular regulatory cells (T$_{FR}$, cluster 11 in unstimulated cells) which, in addition to the T$_{FH}$ signature, also featured a distinct Treg transcriptional profile as well as the upregulation of *POU2AF1* and *CCR9*, was not altered by iLD-IL-2 treatment (Supplementary Fig. 6a).

Next, we compared the transcriptional profile of the four identified T$_{FH}$-like clusters marked by the specific upregulation of IL-21 expression. We identified a continuum of T$_{FH}$ cell differentiation among these clusters, with the CD25$^+$IL-21$^+$ T$_{FH}$ subset representing the terminal stage of maturation in blood, as illustrated by their classical T$_{FH}$ profile and increased IL-21 production (Fig. 3a). Analysis of the co-expression of CXCR5 and IL-21 revealed that IL-21 was preferentially expressed within the CXCR5$^+$ cells (Fig. 3b). The exception was in the CXCR5$^{low}$ IL-21$^+$ cluster, where approximately 50% of the IL-21$^+$ cells were also CXCR5$^{low}$. These data suggest that our single-cell multiomics approach allows us to specifically identify the more differentiated IL-21-producing T cells in this cluster, with a phenotype resembling the CXCR5$^{low}$ PD-1$^{hi}$ circulating T peripheral helper (T$_{PH}$) cells that have been previously shown to be increased in T1D patients[35]. Furthermore, we found similar profiles between the CXCR5$^+$ T$_{CM}$ and the CD25$^+$ T$_{FH}$ clusters sorted from the CD25$^{-/low}$ Tconv and CD25$^+$ Treg gates, respectively, with the latter likely representing an earlier stage of T$_{FH}$-cell differentiation. In agreement with this shared identity and developmental stage, the CXCR5$^+$ T$_{CM}$ and CD25$^+$ T$_{FH}$ subsets displayed largely overlapping embeddings among all stimulated CD4$^+$ T cells (Fig. 3c, d). In contrast to these three conventional T$_{FH}$-like subsets, which seem to share a similar differentiation pathway, the CXCR5$^{low}$IL-21$^+$ CD25$^{-/low}$ Teff cluster represented a much more heterogeneous cluster with a scattered clustering profile and increased IL-21 production (Fig. 3c, d).

We next sought to replicate this LD-IL-2 driven reduction of IL-21$^+$ cells, by delineating the frequency of circulating T$_{FH}$ cells (defined as CD4$^+$CD45RA$^-$CXCR5$^+$PD-1$^+$ T cells; Supplementary Fig. 12a) using a FACS-based strategy. In agreement with our single-cell multiomics data, we observed a small decrease of T$_{FH}$ cells at Day 27 in the 18 DILfrequency participants treated with the 3-day interval dosing schedule (Supplementary Fig. 12b). We also replicated these results in a subset of five DILfrequency patients using a specific intracellular immunostaining panel in PBMCs from four visits (Days 0, 3, 27 and 55), where we observed a consistent 6.3% decrease in the frequency of circulating T$_{FH}$ cells at Day 27 (Supplementary Fig. 12c). These small differences do not reach statistical significance, highlighting the challenge of robustly delineating circulating T$_{FH}$ cells in T1D patients. Furthermore, it also underscores the value of the single-cell multiomics approach, together with in vitro activation, to discriminate the heterogeneity of this subset of cells, and to specifically identify the more differentiated IL-21$^+$ cells, which would not be possible using conventional FACS-based approaches.

To better discriminate the effect of IL-2 on the differentiation of IL-21-producing cells, we then stratified the cells in these four T clusters according to their expression of IL-21 to define activated T$_{FH}$ profiles (Fig. 3e). Consistent with the results from the differential abundance analysis in stimulated cells (Supplementary Figs. 6b, 7e), we found evidence for a reduction in the relative proportion of IL-21$^+$ cells at Day 27 in the two subsets with higher frequencies of IL-21$^+$ cells (Fig. 3f). Furthermore, pseudo-time analysis in total stimulated CD4$^+$ T cells revealed a similar trajectory of T$_{FH}$ cell differentiation characterised by the acquisition of an activated T$_{FH}$ phenotype and production of IL-21 (Supplementary Fig. 8e–g). In contrast with the FOXP3$^+$ Treg trajectory, iLD-IL-2 was found to impair this differentiation trajectory, as illustrated by the reduced pseudo-time distribution at Day 27 among cells from the more differentiated CD25$^+$IL-21$^+$ T$_{FH}$ subset (Supplementary Fig. 8h). These results further support a broader, previously uncharacterized, effect of iLD-IL-2 immunotherapy in potentially inhibiting the differentiation of IL-21-producing cells. In addition to the reduction of IL-21-producing cells, we also obtained some evidence (FDR-adjusted $P = 0.096$) for the reduction of another subset marked by the expression of many pro-inflammatory cytokines, including GM-CSF, IL-21 and IL-2, as well as the classical type-2 inflammatory cytokines IL-4 and IL-13 (CD25$^+$ T$_{H2}$ T$_{EM}$, cluster 6; Supplementary Figs. 4b, 6b).

## Effects of iLD-IL-2 on CD8$^+$ T and CD56$^+$ NK cells

In the CD8$^+$ T cell compartment, differential abundance analysis revealed no discernible differences in the relative composition of the conventional αβ CD8$^+$ T cell subsets after IL-2 treatment. However, we identified a previously uncharacterised reduction of two subsets of innate-like CD8$^+$ T cells, including mucosal-associated invariant T cells (MAIT; cluster 4) and V$_{γ9}$V$_{δ2}$ T cells (cluster 9; Fig. 4a, b and Supplementary Fig. 13). In contrast to the IL-2 induced alterations in the CD4$^+$ T cell compartment, the reduction of circulating MAIT and V$_{γ9}$V$_{δ2}$ T cells was more long-lived and there was still evidence of a reduced frequency in blood one month after the last dose of IL-2 (Fig. 4c).

Finally, in the CD56$^+$ NK cell compartment, despite the lower level of functional heterogeneity and differentiation (Fig. 4d, e), we found that the increased frequency of CD56$^{br}$ NK cells after IL-2 treatment, was driven particularly by an increased relative proportion of an activated HLA-II$^+$ subset (Fig. 4f). By contrast we found no alterations in the relative composition of CD56$^{dim}$ subsets (Fig. 4g and Supplementary Fig. 14). Consistent with the activation phenotype and increased expression of CD56 on CD56$^{br}$ NK cells after treatment, we observed a reduced frequency of contaminating CD56$^{dim}$ and CD56$^{br}$ NK cells sorted from the CD56$^{br}$ and CD56$^{dim}$ NK cell gates, respectively at Day 27 (Fig. 4f, g). The alterations in the CD56$^{br}$ NK cells were also transient

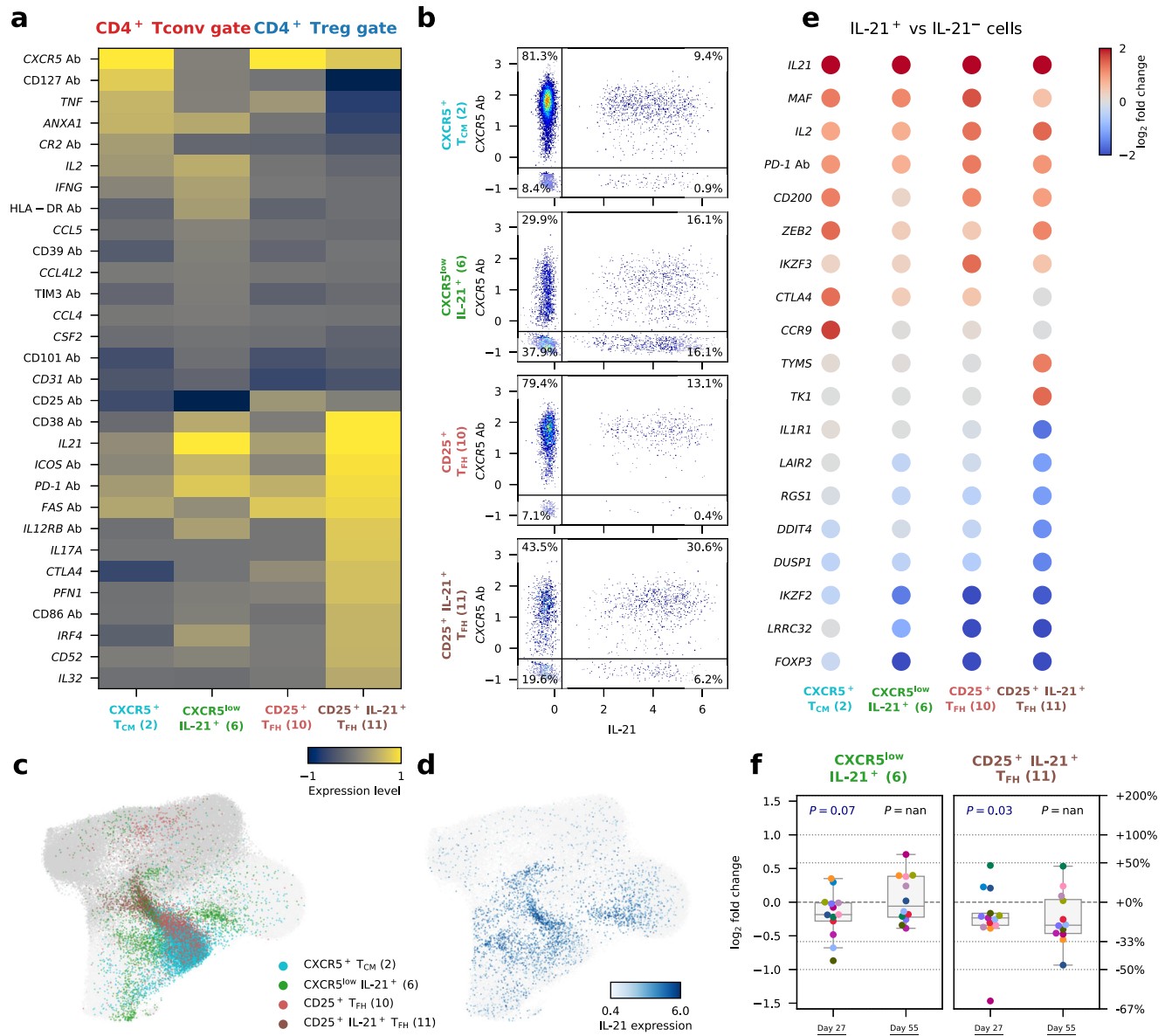

**Fig. 3 | Low-dose IL-2 treatment inhibits the differentiation of IL-21-producing CD4⁺ T cells. a** Expression levels of 30 differentially expressed markers in the four assessed IL-21-producing CD4⁺ T cell subsets from in vitro stimulated CD4⁺ T cells: CXCR5⁺ $T_{CM}$ and CXCR5^low^IL-21⁺ sorted from the CD25^−/low^ Tconv gate; and CD25⁺ $T_{FH}$ and CD25⁺IL-21⁺ $T_{FH}$ sorted from the CD127^low^CD25^hi^ Treg gate. **b** Expression of CXCR5 (AbSeq) and *IL21* in the four assessed IL-21⁺ CD4⁺ T cell subsets. **c, d** UMAP plots depicting the clustering of in vitro stimulated CD4⁺ T cells (*n* = 209,543 cells). Cells are coloured according to the respective IL-21⁺ cluster assignment (**c**); or by normalised and scaled IL-21 expression levels (**d**). Dark and light grey denote cells sorted from the CD127^low^CD25^hi^ Treg gate and CD25^−/low^ Tconv gate, respectively, that are not in IL-21⁺ clusters. **e** Gene expression changes between IL-21⁺ and IL-21⁻ cells within each of the four assessed IL-21⁺ CD4⁺ T cell clusters. Genes with absolute log₂ fold change >= 1.2 and FDR-adjusted *P* values <0.01 in at least one cluster were

selected. Fold change and *P* values were calculated using a generalized linear model implemented in DESeq2 (see "Methods"). **f** Relative frequency changes of IL-21⁺ cells within the two CD4⁺ T cell subsets displaying higher expression levels of IL-21: the CXCR5^low^IL-21⁺ subset (left panel) and the CD25⁺IL-21⁺ $T_{FH}$ subset (right panel), comparing Day 27 and Day 55 with Day 0, respectively. Each dot represents cells from a single participant. Dots with the same colour represent the same participant. *n* = 13 participants. Each box ranges from the first quartile (Q1) to the third quartile (Q3), with a central line indicating the median. The bottom and top whiskers extend to the most extreme data points within Q1−1.5 × (Q3 − Q1) and Q3 + 1.5 × (Q3 − Q1), respectively. *P* values were calculated by comparing the relative frequency values between the two time points using a two-sided Wilcoxon signed-rank test. Source data for (**a**, **e**, **f**) are provided as a Source Data file.

and were not maintained one month after treatment (Supplementary Fig. 14b, c).

### Interval low-dose IL-2 immunotherapy induces a systemic and long-lasting anti-inflammatory gene expression signature

We next investigated whether, in addition to cellular composition alterations, we could detect IL-2-induced gene expression changes. On Day 27, we identified 40 genes that are differentially expressed in one

or more subsets of unstimulated and stimulated cells (Fig. 5a–c). Consistent with previous flow cytometry analyses[21], CD25 was strongly upregulated on Tregs on Day 27. The differentially expressed genes were mostly restricted to the more IL-2-sensitive CD127^low^CD25^hi^ Treg and CD56^br^ NK cell subsets and could be broadly categorised into CD4⁺ Treg and CD56^br^ NK cell signature genes, reflecting the relative increased proportion of both subsets after iLD-IL-2 treatment, as well as the identified reduction in actively dividing CD56^br^ NK cells

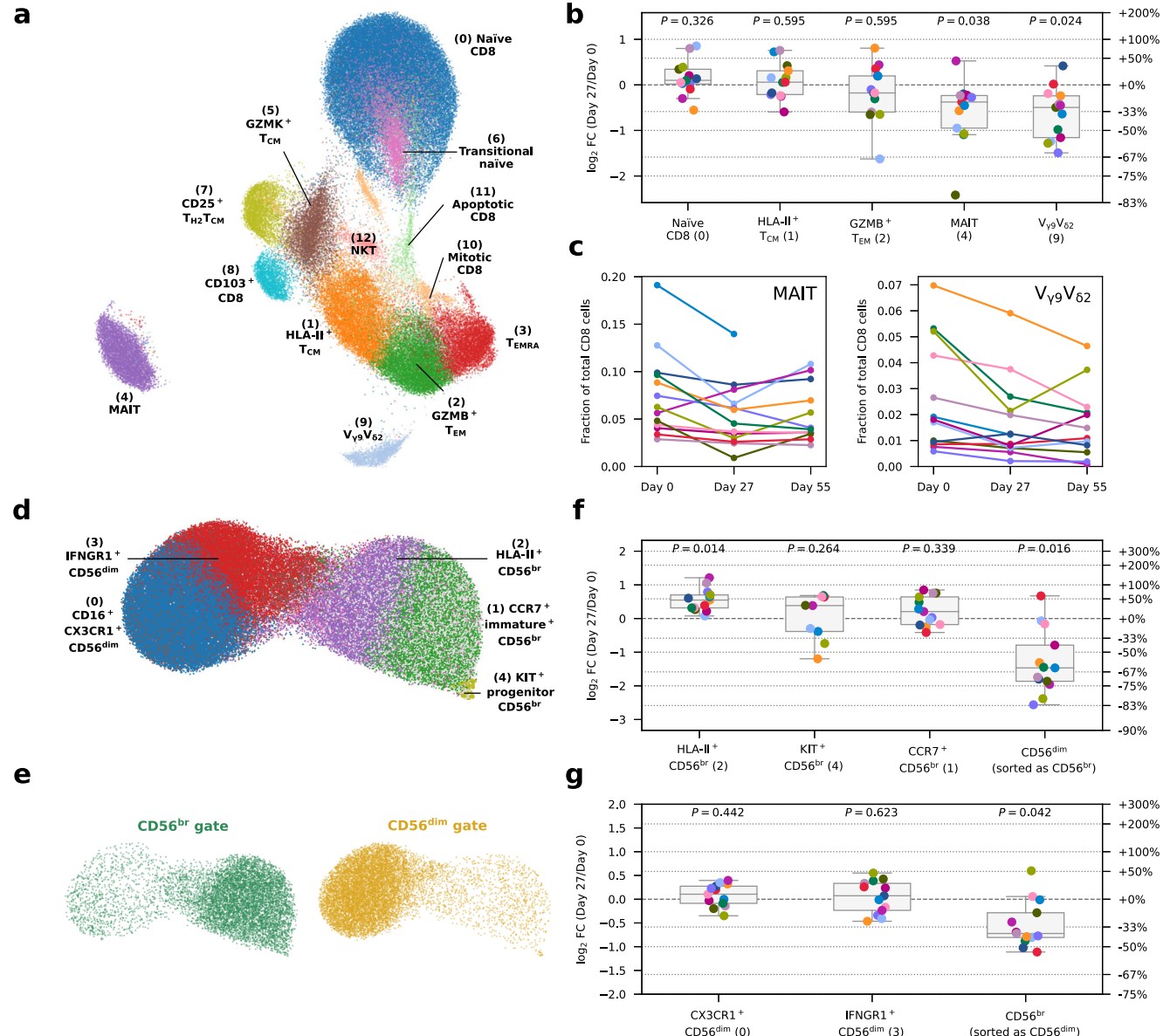

**Fig. 4 | IL-2 treatment reduces the frequency of innate-like CD8⁺ MAIT and V_γ9V_δ2 T cells in circulation and selectively expands the IL-2 sensitive CD56^br NK cell subset. a** UMAP plot depicting the clustering of $n = 93,379$ unstimulated CD8⁺ T cells. Clusters were manually annotated based on the expression of key mRNA and protein markers. **b** Relative frequency changes of five selected CD8⁺ T cell clusters on Day 27 compared to the baseline (Day 0) levels, for unstimulated cells. **c** Relative frequency of the innate-like mucosal-associated invariant T cell (MAIT; left panel) and V_γ9V_δ2 T cell (right panel) clusters on Day 0, Day 27, and Day 55. Each line represents cells from a participant. Lines with the same colour represent the same participant. $n = 13$ participants. **d** UMAP plot depicting the clustering of $n = 118,625$ unstimulated NK cells. Clusters were manually annotated based on the expression of key mRNA and protein markers. **e** UMAP plot as in (**d**), split and coloured by FACS sorting gates of origin. **f, g** Relative frequency changes of five selected CD56^br (**f**) or CD56^dim (**g**) NK cell clusters on Day 27 compared to the baseline pre-treatment (Day 0) levels, for unstimulated cells. The relative frequency of each cluster is calculated as the number of cells in that cluster divided by the total number of cells sorted from the CD8 (**b**), CD56^br (**f**) or CD56^dim (**g**) NK cell gate. Each dot represents cells from a single participant. Dots with the same colour represent the same participant. $n = 13$ participants. Each box ranges from the first quartile (Q1) to the third quartile (Q3), with a central line indicating the median. The bottom and top whiskers extend to the most extreme data points within $Q1 − 1.5 \times (Q3 − Q1)$ and $Q3 + 1.5 \times (Q3 − Q1)$, respectively. P values were calculated using two-sided Wilcoxon signed-rank tests based on cell population frequencies between Day 27 and Day 0, followed by Benjamini–Hochberg FDR adjustment. Source data for (**b, c, f, g**) are provided as a Source Data file.

(Supplementary Fig. 3b). Cycling cells were not present in the stimulated cell populations, potentially due to the preferential sensitivity of cycling cells to apoptosis following activation.

Contrary to the differential expression on Day 27, which was observed exclusively in the IL-2-sensitive Treg and CD56^br subsets, differential expression on Day 55, one month after the last IL-2 injection, was widespread and displayed a consistent pattern in all five assessed immune subsets (Fig. 6a and Supplementary Fig. 15a, b). These findings indicate that our iLD-IL-2 dosing regimen induces a shared Day 55 gene expression signature in different cell types, which featured most prominently the upregulation of Cytokine Inducible SH2 Containing Protein (*CISH*), a gene encoding a well-characterised negative regulator of cytokine signalling, and the downregulation of Amphiregulin (*AREG*), a secreted TNF-inducible protein with pleiotropic roles in inflammation[36]. We also noted an enrichment of genes associated with cytokine signalling, most notably in the tumour necrosis factor (TNF) signalling pathway, which in addition to *CISH* and *AREG*, also included *TNFSF14, TNFSF10, STAT1, SGK1, NFKBIZ, NFKBIA,*

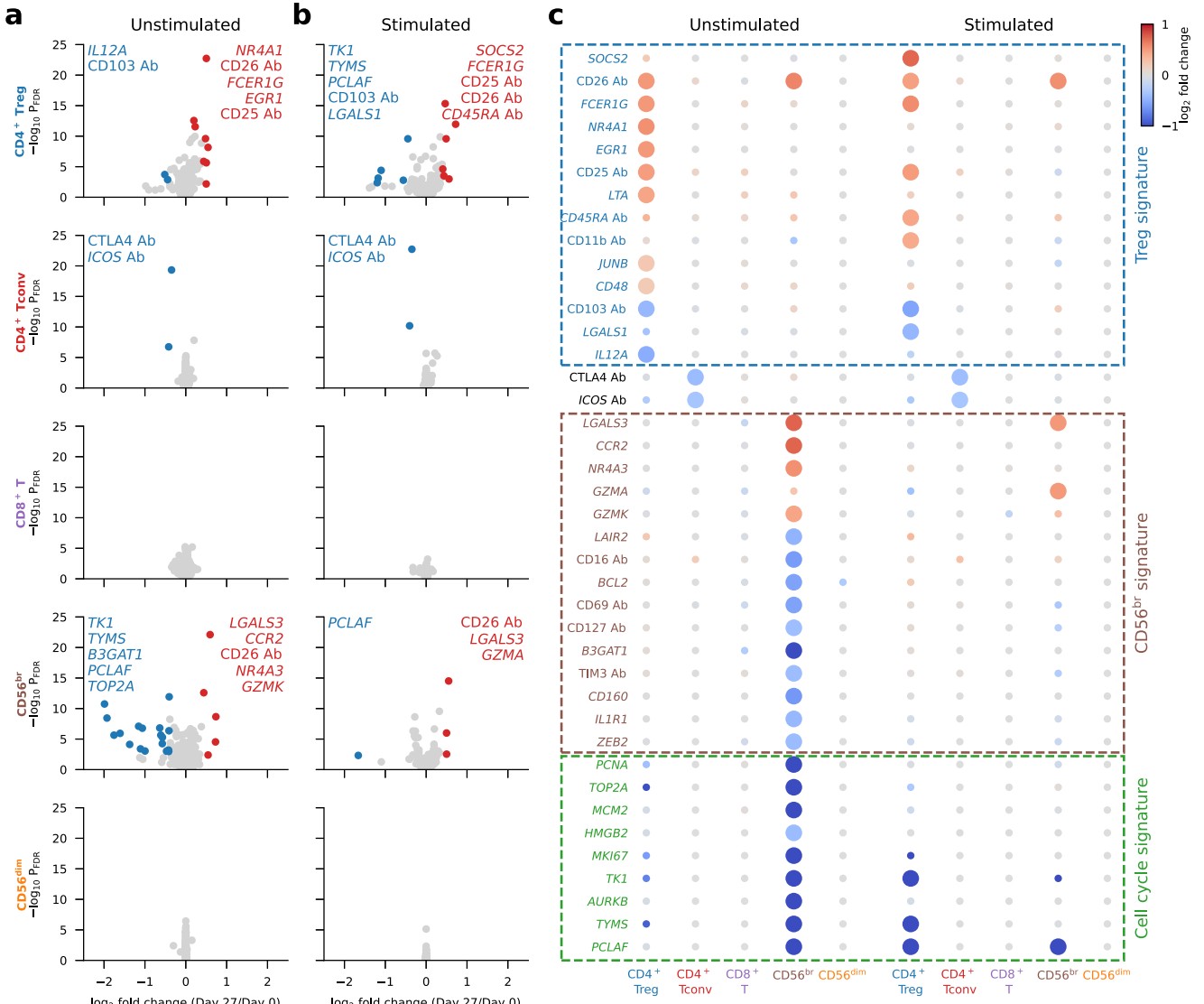

**Fig. 5 | IL-2 treatment selectively modulates gene expression of CD25hi Tregs and CD56br NK cells at Day 27. a, b** Volcano plots depicting gene expression changes between Day 0 and Day 27 for the five assessed immune populations, for unstimulated (**a**) and stimulated (**b**) cells. Significantly differentially expressed genes are coloured in red (upregulated genes) or blue (downregulated genes). Names of the top five up- and downregulated genes as defined by fold change are labelled on each panel. **c** Differential expression of Day 27 signature genes. Day 27 signature genes are defined as the 40 genes that are significantly differentially expressed in at least one cell population comparing Day 27 with Day 0. Dot colours represent log₂ fold change between Day 0 and Day 27. Larger dots represent genes that are significantly differentially expressed in the respective cell population. Manually annotated signature gene subsets are labelled with dashed boxes. In (**a–c**), fold change and *P* values were calculated using a generalized linear model implemented in DESeq2 (see Methods). Source data for (**a**, **b**, **c**) are provided as a Source Data file.

*DUSP2*, *DUSP4*, *DUSP5*, *RGS1* and *TNFAIP3* (Fig. 6a). Furthermore, we observed a downregulation of several TNF-inducible genes that play a central role in curtailing TNF signalling, including the inhibitors of NFκB, *NFKBIZ* and *NFKBIA*, *RGS1*, *TNFAIP3* and *AREG*. Consistent with this observation, Oncostatin M (*OSM*), a pro-inflammatory cytokine shown to be increased in inflammatory bowel disease patients with poor response to anti-TNF therapy[37], was also downregulated by iLD-IL-2. Increased expression of *TNFSF14*, which encodes LIGHT, is notable since decreased expression is associated with higher susceptibility to multiple sclerosis[38] and in a mouse model of multiple sclerosis, LIGHT expression in the brain limits disease severity[39]. The widespread detection of this signature in all five T and NK cell populations assessed in this study suggests that this effect is not driven by IL-2 signalling directly but is likely the result of the sustained cellular changes induced by iLD-IL-2 immunotherapy.

Although this Day 55 expression signature was detected in 11 of the 12 participants with Day 55 data, we observed heterogeneity among participants for the expression levels of the same genes on Day 27 (Fig. 6b). Some participants exhibited concordant expression changes on Day 27 and Day 55, compared to Day 0, while others showed discordant expression profiles, with *CISH* being downregulated and *AREG* upregulated on Day 27. Moreover, by principal component analysis (PCA), we found that this concordant-discordant axis, captured by the first principal component (PC1), was able to largely explain the inter-individual heterogeneity of Day 55 signature genes on Day 27 and appeared to be associated with IL-2 dose (Supplementary Fig. 15c). Considering this, we derived a linear score to quantify the changes in the Day 55 signature for each participant, referred to as the Day 55 signature score, with higher scores reflecting higher expression of signature genes upregulated on Day 55 (e.g. *CISH*), and lower

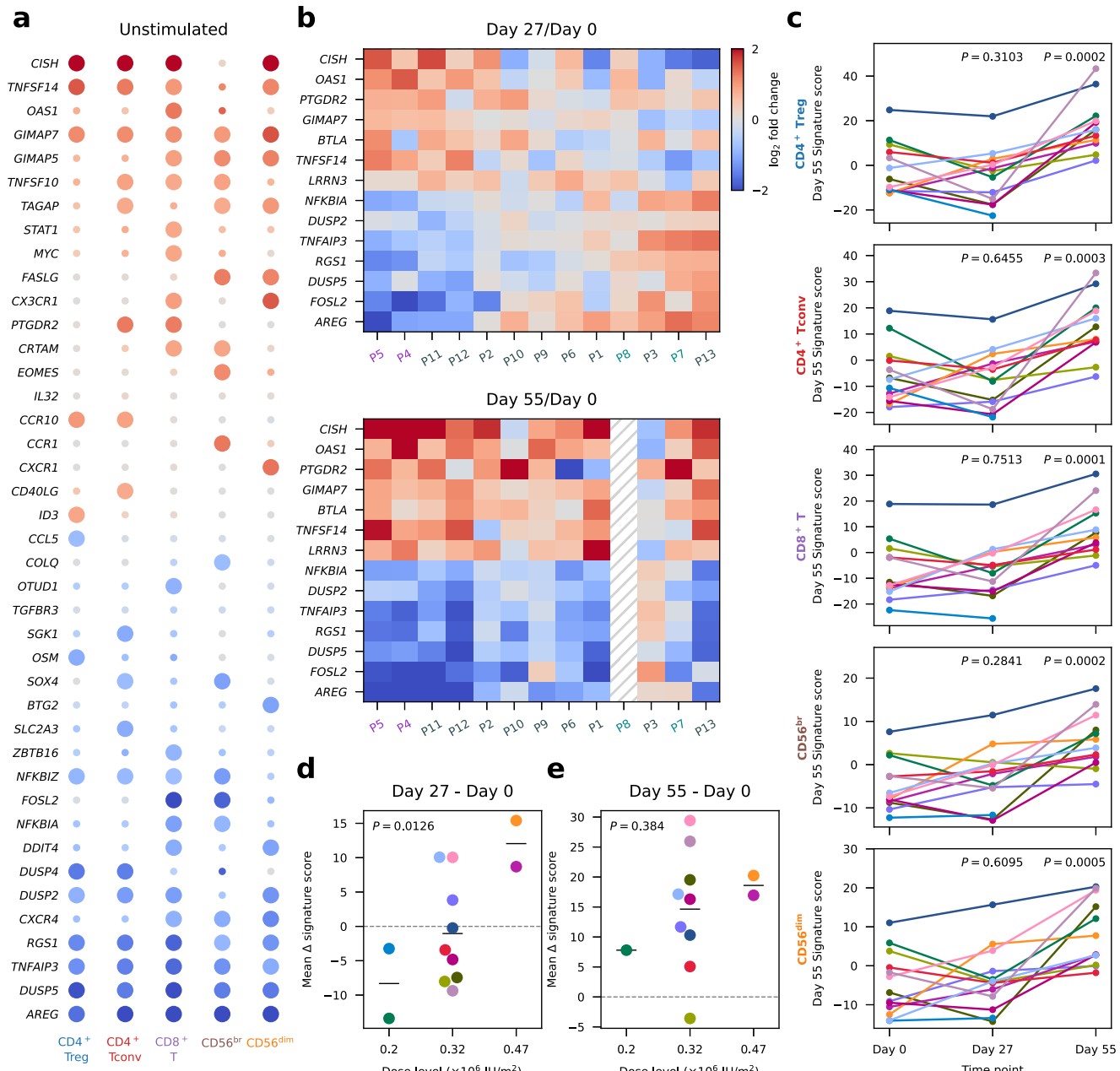

**Fig. 6 | Interval low-dose IL-2 immunotherapy induces a long-lasting anti-inflammatory gene expression signature. a** Differential expression of Day 55 signature genes. Day 55 signature genes were defined as the 41 genes significantly differentially expressed in at least one cell population comparing Day 55 with Day 0. Dot colours represent log₂ fold-change between Day 0 and Day 55. Larger dots represent significantly differentially expressed genes. Fold change and *P* values were calculated using a generalized linear model implemented in DESeq2 (see "Methods"). **b** Participant-specific differential expression in CD8⁺ T cells comparing Day 27 (top panel) or Day 55 (bottom panel) with Day 0. Top 5 up- and downregulated genes as defined by Day 55/Day 0 fold-change in CD8⁺ T cells. Participant labels are coloured by their respective IL-2 dose (×10⁶ IU/m²): 0.2 (cyan),

0.32 (grey), 0.47 (purple). **c** Day 55 signature scores by participant for each time point and cell type. Each participant is represented by a different colour. *P* values were calculated using a two-tailed paired *t* test comparing Day 27 or Day 55 with Day 0, respectively. **d, e** Variation in the mean expression scores of Day 55 signature genes across the five major cell types (mean Δ signature score) between Day 27 and Day 0 (**d**) and between Day 55 and Day 0 (**e**) for each participant. Participants were stratified by IL-2 dose. *P* values were calculated using linear regression, without multiple-comparison adjustments. In (**d** and **e**), average values of the Day 55 signature in all five cell types are shown for each participant. *n* = 13 participants. Source data for (**a**–**e**) are provided as a Source Data file.

expression of signature genes downregulated on Day 55 (e.g. *AREG*). As expected, the Day 55 signature score showed considerable inter-individual variation (Fig. 6c) that was consistent across different cell types (Supplementary Fig. 15d). This enabled us to correlate changes in signature score, which indicated the direction and amplitude of the modulation of Day 55 signature genes, with IL-2 dose level. We confirmed that participants receiving higher doses of IL-2 were more likely to have increased signature scores on Day 27 and Day 55 compared the

baseline, indicating concordant expression changes, whilst participants receiving lower doses tended to show discordant changes (Fig. 6d, e). The dose-dependence of the Day 55 signature reached statistical significance only when comparing to the participant-specific baseline, but not when inspecting the cross-sectional data at each time point (Supplementary Fig. 15e). This underscores the importance of our longitudinal study design that enhanced our statistical power to identify differential expression patterns.

## Discussion

Our findings not only extend previous studies[17–19,26] but also provide insight into the mechanism of action of LD-IL-2 that has practical implications for its clinical use: firstly, we show that LD-IL-2 administered every three days for a period of one month is able to selectively increase the frequency and number of thymic-derived FOXP3+HELIOS+ CD4+ Tregs in all 18 study participants analysed, while showing no detectable impact on the expansion or activation of conventional effector T or CD56dim NK cells. The selective expansion of HELIOS+ Tregs and CD56br NK cells is consistent with previous findings in chronic graft-versus-host disease patients treated with daily dosing of LD-IL-2, using a mass cytometry approach[40], demonstrating the high sensitivity of these immune subsets to LD-IL-2 treatment. We observed a particularly pronounced increase in the frequency of naïve FOXP3+HELIOS+ Tregs after iLD-IL-2 therapy, indicating that the Treg increases are caused by an increased representation of thymic-derived Tregs, and not the proliferation of peripherally-induced Tregs. These findings are consistent with the increased clonal diversity in T cells following an escalating LD-IL-2 regimen in graft-versus-host disease patients[41,42]. Secondly, we also noted that our iLD-IL-2 schedule did not induce a detectable cytotoxic gene expression signature in any T cell subset, indicating that among T cells, iLD-IL-2 has a Treg-specific effect and does not lead to the activation of a pro-inflammatory response. In addition to Tregs, the CD56br NK cell population is probably also contributing to the overall anti-inflammatory effects of iLD-IL-2, in an immunoregulatory role, as previously described[43,44]. Thirdly, iLD-IL-2 was able to reduce IL-21-producing T cells, possibly by inhibiting their differentiation. IL-2 signalling has been shown to inhibit the differentiation of T$_{FH}$ cells in vivo in mice[45], and in an in vitro model in humans[46]. Consistent with these findings, LD-IL-2 has been shown to decrease the frequency of T$_{FH}$ cells, as defined by cell surface markers[47], as well as the frequency of marginal zone B cells[6] in SLE patients, supporting a role in the regulation of germinal centre reactions. Our results extend these findings to T1D and demonstrate a specific effect of iLD-IL-2 treatment in reducing pathogenic IL-21-producing CXCR5+ T$_{FH}$ cells as well as IL-21-producing CXCR5low T cells. Previously, we have shown that a two-fold increase in IL-2, but not IL-21, was responsible for the protective effect conferred by the *Idd3* locus (containing *IL2* and *IL21*) in the nonobese diabetic (NOD) mouse model of T1D[15], and independently that a reduction in IL-21 correlated with increased IL-2[48]. In humans, IL-21 has been implicated in the pathogenesis of several autoimmune diseases[49,50], particularly T1D[35,51,52]. Together, these observations lend further support for the use of iLD-IL-2 in T1D as well as in other diseases associated with increased IL-21 production, such as systemic lupus erythematosus, rheumatoid arthritis, and psoriasis. In support of this therapeutic strategy, an anti-IL-21 monoclonal antibody in combination with liraglutide showed promising results in preserving beta-cell function[53]. Of note, we observed no evidence for the alteration of the frequency of FOXP3+ T$_{FR}$ cells, which has been previously shown to play a role in the regulation of T$_{FH}$ cells and the germinal centre reaction[54,55].

We also observed a decrease in circulating innate-like CD8+ MAIT and V$_{γ9}$V$_{δ2}$ T cell subsets during treatment and one month after. Since these cells function in anti-viral and anti-bacterial defence[56], our findings suggest a role of IL-2 immunotherapy in the recruitment of MAIT and V$_{γ9}$V$_{δ2}$ T cells to tissues following treatment, thereby increasing defence against viral and bacterial infections[57]. This could provide a mechanism for the observed decreased incidence of viral infections in SLE patients undergoing LD-IL-2 immunotherapy[58].

The cellular alterations induced by iLD-IL-2 treatment led to the establishment of a long-lived gene expression signature in all assessed immune populations, which suggest a novel immunoregulatory mechanism of iLD-IL-2. Although the effects of this signature are not yet understood, it is marked by the distinct upregulation of *CISH*, a well-characterised negative regulator of cytokine signalling whose expression is known to be induced by IL-2, as well as other cytokines, through the activation of STAT5[59]. In agreement with this central role in immune signalling, CISH has been recently shown to inhibit the polarization of naïve T cells into the Th2 and Th9 lineages[60]. Together with our data, these findings suggest that sustained increased expression of CISH in response to iLD-IL2 may alter the cytokine signalling threshold required for T cell activation and acquisition of effector functions and could inhibit the differentiation of pathogenic T cells. In contrast, among the downregulated genes, we observed several negative regulators of an inflammatory response at Day 55, including key TNF-inducible genes such as *TNFAIP3*, *RGS1* and *AREG*, suggesting that iLD-IL-2 can decrease the homoeostatic levels of TNF for at least one month after the cessation of dosing. Note also that both *RGS1* and *TNFAIP3* are established candidate susceptibility genes for T1D and several other autoimmune diseases[61].

One possible explanation for the longevity of the signature is that TNF and IL-2 are among the cytokines known to bind to the extracellular matrix[30–33] and that the retention of the cytokines within tissues extends far beyond their initial increases[62]. The establishment of the Day 55 IL-2 signature is dose-dependent at Day 27, likely reflecting a dose-dependent accumulation of injected IL-2 on the extracellular matrix within the interstitial space following subcutaneous dosing. We hypothesise that the balance between pro- and anti-inflammatory cytokines bound to the extracellular matrix provides a regulatory mechanism to control immune responses. Circulating immune cells can constantly integrate this information through a combination of cell-type specific receptors. Under homoeostatic conditions, this provides a dynamic buffer system preventing excessive (or potentially chronic) immune activation. A perturbation of this balance via either sustained regulatory (e.g. iLD-IL-2 immunotherapy) or inflammatory (e.g. infection) environment can alter the composition of the soluble mediators contained in the extracellular matrix, and consequently affect the interaction between the immune system and the local environment, leading to the remodelling of their transcriptional profile and an alteration of their threshold for further stimulation. Such a complex system is likely critical not only to prolong the biological activity of key immune mediators such as TNF and IL-2, but also to control site specificity to prevent improper signalling in the wrong physiological context.

A limitation of this study is that we were not able to elucidate the exact mechanism underpinning the induction of the identified long-lived gene expression signature and how this is regulated by iLD-IL-2. Although we show that iLD-IL-2 can induce this signature, it is certainly not the only responsible factor. Furthermore, markers of disease activity and progression are limited in T1D and were not collected for the DILfrequency patients, which precludes the analysis of how they are associated with these gene expression alterations. A more comprehensive investigation of these questions will need to be performed in an appropriate follow up cohort such as the recently completed Interleukin-2 Therapy of Autoimmunity in Diabetes (ITAD) trial[63]. The findings from this study will inform the design of suitable follow up experiments in this and other studies of iLD-IL-2 in multiple diseases and will be critical to better understand both the longevity of the signature and how it is associated with clinical outcomes. Another limitation is that the interpretation of the results may be dependent on our specific iLD-IL-2 dosing regimen. Although it is likely that most regimens will induce very similar results, it will be important to validate these findings in different dosing schedules.

Taken together, our findings provide a better understanding of the mechanism of action of LD-IL-2 immunotherapy. In the past there have been concerns over the very short half-life of recombinant IL-2 in vivo and that NK cell expansion might be detrimental, which have led to the development of genetically-modified IL-2 muteins with increased half-life and preferential binding to the high-affinity trimeric IL-2 receptor in order to reduce CD56br NK cell stimulation. Our results

demonstrating a long-lived gene expression signature associated with potential anti-inflammatory effects, detectable one month after the end of aldesleukin treatment, alleviate some of these concerns and support the therapeutic potential of IL-2 that is not engineered for longer in vivo half-life or preferential binding to the high affinity IL-2 receptor, given intermittently and at low doses. Given the diverse range of conditions where LD-IL-2 immunotherapy is being trialled, it is critical to better understand how dosing regimen affects the drug's mechanism of action, and to tailor it to the disease-specific applications in the context of standard of care[64]. In T1D, clinical efficacy trials are ongoing with LD-IL-2 to reduce the decline of C-peptide in newly-diagnosed patients (ITAD[63] and DIABIL-2; ClinicalTrials.gov identifiers NCT03782636 and NCT02411253, respectively). Another possibility is that LD-IL-2 treatment could aid in the maintenance of the clinical benefits observed in T1D with anti-CD3 (teplizumab) and anti-TNF (golimumab)[65]. Our study adds to the good track record of using low-dose recombinant IL-2 and confirms its very strong safety profile, which is critical for the potential long-term use of LD-IL-2 immunotherapy for diseases such as T1D in which the target patient population will consist mainly of children, including the possible future use of iLD-IL-2 treatment in the prevention of T1D diagnosis.

## Methods

The study was performed in accordance with the guidelines for good clinical practice and the Declaration of Helsinki. Study participants included T1D patients enroled in two single centre mechanistic studies of low-dose recombinant IL-2 (aldesleukin) immunotherapy: (i) the Adaptive study of IL-2 dose on regulatory T cells in type 1 diabetes (DILT1D), a non-randomised single IL-2 dosing study[20]; and (ii) the Adaptive study of IL-2 dose frequency on regulatory T cells in type 1 diabetes (DILfrequency), a response-adaptive trial of repeated doses of IL-2 administered using an interval dosing approach[21]. Approval was obtained from the Health Research Authority, National Research Ethics Service (13/EE/0020 and 14/EE/1057, for the DILT1D and DILfrequency studies, respectively), London, United Kingdom. The trials were registered at the International Standard Randomised Controlled Trial Number Register (ISRCTN27852285 and ISRCTN40319192) and ClinicalTrials.gov (NCT01827735 and NCT02265809). The study protocols were published in advance of the completion and final analysis of the trials[66,67]. All participants provided written informed consent prior to their participation in the studies.

### Study design and participants

Study participants comprised 18 T1D patients enroled in the DILfrequency study[21]. All 18 study participants, who were all adult patients diagnosed with T1D over 60 months before recruitment into the study, were selected from the 3-day interval dosing group and were treated with IL-2 with doses ranging from 0.2 to $0.47 \times 10^6$ IU/m².

FACS-based replication of the single-cell multiomics findings was performed in two follow-up cohorts: (i) a single-dosing cohort, comprising 6 DILT1D patients[20] treated with a single dose of IL-2 ranging from 0.6 to $1.5 \times 10^6$ IU/m²; and (ii) a multiple dosing cohort, comprising 5 DILfrequency patients from the three-day dosing regimen, treated with IL-2 doses ranging from 0.2 to $0.32 \times 10^6$ IU/m². Study participants' characteristics and immune profiling at baseline are summarised in Supplementary Data 1.

### Flow cytometry

Flow cytometric characterisation of fresh whole-blood samples from the DILfrequency study participants was performed at each visit within 4 h of phlebotomy. Surface immunostaining was performed by incubating 150 µl whole blood for 45 min at room temperature with fluorochrome-conjugated antibodies (Supplementary Data 2) with Brilliant Stain buffer (BD Biosciences). Red blood cells were then lysed (BD FACS Lysing solution, BD Biosciences). The absolute number of

the main lymphocyte populations, including CD4+ T, CD8+ T and CD56+ NK cells, was assessed using a whole-blood BD Multitest 6-Colour TBNK assay using BD Trucount Tubes according to the manufacturers' instructions (BD Biosciences) at each visit. Absolute numbers of CD4+ Tregs and Tconvs, and CD56br and CD56dim NK cells were derived by multiplying the absolute number of CD4+T cells and CD56+ NK cells by the appropriate percentages derived from the panel described above.

For the intracellular immunostainings, cryopreserved PBMCs were thawed at 37 °C and resuspended drop-by-drop in X-VIVO 15 (Lonza) with 1% heat-inactivated, filtered human AB serum (Sigma). PBMC samples were selected from 10 visits (Days 0, 1, 2, 3, 4, 5, 6, 9, 14 and 28) from each patient from the single-dosing and 4 visits (Days 0, 3, 27 and 55) for the multiple dosing cohort. PBMCs were then washed in PBS, incubated with Fixable Viability Dye eFluor 780 (eBioscience) for 15 min at 4 °C and stained with fluorochrome-conjugated antibodies against surface-expressed markers (Supplementary Data 2) with Brilliant Stain buffer for 45 min at 4 °C. Fixation and permeabilisation was performed using the FOXP3 Fix/Perm Buffer Set (eBioscience) according to the manufacturer's instructions, and cells were then stained with fluorochrome-conjugated antibodies against intracellular markers (Supplementary Data 2) with Brilliant Stain buffer for 1 h at room temperature.

Immunostained samples were acquired on a BD Fortessa (BD Biosciences) flow cytometer with FACSDiva software (BD Biosciences) and analysed using FlowJo (Tree Star, Inc.). All FACS data was analysed and visualised using GraphPad Prism (v9.3.1). Information for all fluorochrome-conjugated antibody panels used in this study is provided in Supplementary Data 2.

### In vitro Treg suppression assay

A total of 12 DILfrequency participants were selected for this analysis, including nine participants that were also included in the single-cell analyses. For each participant, cells sampled from three time points were used: Day 0, Day 24 and Day 55. Cryopreserved PBMCs were thawed as described above and stained with a cocktail of monoclonal antibodies (Supplementary Data 2) for cell sorting. Suppression assays were established in V-bottom 96-well plates by sorting 500 CD4+CD25−/loCD127+ effector T cells (Teffs) in the presence or absence of CD4+CD25highCD127low Tregs at various ratios (Treg:Teff 0:1, 1:4, 1:2 and 1:1) with $1 \times 10^3$ CD19+ B cells, using the BD FACSAria III flow cytometer automated cell deposition unit (BD Biosciences). Cells were sorted into in 150 µl X-VIVO 15 supplemented with Penicillin-Streptomycin-Fungizone (Gibco) and 10% heat-inactivated, filtered human AB sera, stimulated with PHA (4 µg/ml; Alere) and incubated at 37 °C, 5% CO₂, for 5 days. Proliferation was assessed by the addition of 0.5 µCi/well [3H] thymidine (PerkinElmer) for the final 20 h of co-culture. Conditions were run in 5 replicates, and proliferation readings (counts per minute [CPM]) averaged. Two samples with averaged proliferation less than 1,000 CPM from the Teff wells alone were excluded from the analysis. The percentage suppression in each culture was calculated using the following formula: percent suppression = 100 − [(CPM in the presence of Tregs ÷ CPM in the absence of Tregs) × 100]. All time points from a participant were analysed concurrently.

### Single-cell multiomics: cell preparation and capture

Single-cell multiomics profiling was performed using the BD Rhapsody system on PBMC aliquots from 13 of the DILfrequency participants treated with the 3-day interval dosing schedule (Supplementary Data 1) collected at the baseline visit (Day 0), 3 days after the last dose of IL-2, immediately prior to the last IL-2 infusion of the dosing phase (Day 27) and one month after the last dose of IL-2 (Day 55). Cryopreserved PBMCs were thawed at 37 °C and resuspended drop-by-drop in X-VIVO 15 with 1% heat-inactivated, filtered human AB serum. After washing in PBS, cells were incubated with Fixable Viability Dye eFluor 780 (eBioscience) for 10 min at room temperature, washed with cold

PBS + 2% FBS and incubated for 30 min at 4 °C with fluorochrome-conjugated antibodies for sorting (Supplementary Data 2) with Brilliant Stain buffer. Cells were then washed two times and resuspended in cold PBS + 1% FBS for cell sorting at 4 °C in a BD FACSAria Fusion sorter (BD Biosciences). The following T and NK cell subsets were sorted from each participant at each visit: (i) CD3⁻CD56ʰⁱ (CD56br NK); (ii) CD3⁻CD56⁺ (CD56dim NK); (iii) CD3⁺CD8⁺ (CD8⁺ T); (iv) CD3⁺CD4⁺ CD25⁻/low (CD4⁺ Tconv); and (v) CD3⁺CD4⁺ CD127lowCD25hi (CD4⁺ Treg). We aimed to sort a total of 130,000 cells for the Day 0 and Day 55 visits and 160,000 for the Day 27 visit, with the following proportion of each of the sorted cell populations: 30% CD4⁺ Treg; 25% CD4⁺ Tconv and CD8⁺ T; 12% CD56br NK and 8% CD56dim NK cells.

Following cell sorting, each sorted subset was labelled with the respective sample barcoding antibody (sample multiplexing kit; BD Biosciences) for 20 min at 4 °C to allow for the identification of the sorting gate and visit of each captured single cell. To minimise day-to-day variation, we processed the three PBMC samples corresponding to the three visits from each participant on the same day. As the number of barcoded oligo-conjugated antibodies available in the kit was less than the total number of sorted subsets, we combined the CD4⁺ Treg and CD56dim NK cells as well as the CD4⁺ Tconv and CD56br NK cells from each visit and labelled the pooled cells with the same barcode. These combined subsets were chosen for their highly differentiated transcriptional and proteomics profile, which allowed robust separation of the respective original subsets at the analysis stage. Cells were then washed three times with BD Sample Buffer (BD Biosciences) at 4 °C to remove any residual unbound barcoding antibody and then pooled together for downstream processing.

To profile unstimulated cells, half of the pooled cells volume (corresponding to ~210,000 cells) were initially incubated at 4 °C for 5 min with human Fc block (BD Biosciences) and then for a further 45 min with a master mix of 65 oligo-conjugated AbSeq antibodies (BD Bioscience; Supplementary Data 2). Cells were then washed three times with BD Sample Buffer at 4 °C to remove residual unbound oligo-conjugated AbSeq antibodies, resuspended in 500 μl cold BD Sample buffer and filtered through a sterile 50 μm Filcon cup-type filter (BD Biosciences) for cell counting. Samples were then resuspended in 620 μl of cold BD sample buffer at a concentration of 40 cells/μl - for an estimated capture rate of ~20,000 single-cells / cartridge - and immediately loaded on two BD Rhapsody cartridges (BD Biosciences) for single-cell capture.

For in vitro stimulation, the remaining volume from the pooled cells was washed at 4 °C and resuspended in X-VIVO 15 + 10% heat-inactivated, filtered human AB serum at a final concentration of $2 \times 10^6$ cells/ml. A volume of 100 μl (~200,000 cells) was then incubated in one well of a round-bottom 96-well plate at 37 °C for 90 minutes with a PMA and ionomycin cell stimulation cocktail (eBioscience), in the absence of protein transport inhibitors. Cells were harvested into a FACS tube, washed with PBS + 2% FBS and processed for cell capture, as detailed above for the unstimulated cells.

## Single-cell multiomics: cDNA library preparation and sequencing

Single-cell capture, and cDNA library preparation was performed using the BD Rhapsody Express single-cell analysis system (BD Biosciences), according to the manufacturer's instructions. cDNA was initially amplified for 11 cycles (PCR1) using the pre-designed Human T-cell Expression primer panel (BD Biosciences) containing 259 primer pairs, together with a custom designed primer panel containing 306 primer pairs (BD Biosciences). A total of 565 primer pairs targeting 534 different genes were used in this study (Supplementary Data 2). To increase the clustering resolution, our targeted panel was enriched for highly variable genes that contribute highly to the clustering of the identified T and NK cell subsets, as well as lineage-defining

transcription factors and other key functional markers of Th cell differentiation to increase clustering resolution.

The resulting PCR1 products were purified using AMPure XP magnetic beads (Beckman Coulter) and the respective mRNA and AbSeq/sample tag products were separated based on size selection, using different bead ratios (0.7× and 1.2×, respectively). The purified mRNA and sample tag PCR1 products were further amplified (10 cycles) using a nested PCR approach and the resulting PCR2 products purified by size selection (0.8× and 1.2× for the mRNA and sample tag libraries, respectively). The concentration, size and integrity of the resulting PCR products were assessed using both Qubit (High Sensitivity dsDNA kit; Thermo Fisher) and the Agilent 4200 Tapestation system (Agilent). The final products were normalised to 2.5 ng/μl (mRNA), 1.1 ng/μl (sample tag) and 0.35 ng/μl (AbSeq) and underwent a final round of amplification (six cycles for mRNA and sample tag and 7 cycles for AbSeq) using indexes for Illumina sequencing. Final libraries were quantified using Qubit and Agilent Tapestation and pooled (~29/67/4% mRNA/AbSeq/sample tag ratio) to achieve a final concentration of 5 nM. Final pooled libraries were spiked with 15% PhiX control DNA to increase sequence complexity and sequenced (150 bp paired-end) on a NovaSeq 6000 sequencer (Illumina).

## Pre-processing of single-cell sequencing data

The single-cell sequencing reads were processed according to the BD Rhapsody data analysis protocol, with slight modifications. First, low-quality read pairs were removed based on read length, mean base quality score and highest single-nucleotide frequency. High-quality R1 reads were analysed to extract cell labels and unique molecular identifier (UMI) sequences. High-quality R2 reads were aligned to the reference panel sequences using Bowtie2[68]. Reads having identical cell labels and UMI sequences and aligned to the same gene were collapsed into a single molecule. To correct sequencing and PCR errors, the obtained counts were adjusted by a recursive substitution error correction (RSEC) algorithm developed by BD Biosciences. The sampling time point and sorting gate of origin of each cell were labelled using information from barcoded oligo-conjugated antibodies. Cells without barcoding information were labelled as "tag N/A", and cells with two or more barcodes were labelled as "multiplet". We note that sample barcoding and AbSeq staining efficiency were compromised in CD56br NK cells. As the quality of the main mRNA library was not affected by this technical issue, CD56br NK cells without barcoding information were included in the quality control and clustering step, but excluded from subsequent analyses dependent on sampling time point (see below). This led to a significant reduction in the number of CD56br NK cells analysed and a concomitant reduction in statistical power to identify IL-2-induced alterations in this subset. Nevertheless, we also note that our cell-enrichment strategy compensated for this cell loss due to technical reasons and provided sufficient resolution to dissect the heterogeneity of the rare population of CD56⁺ NK cells.

## Doublet scoring

Doublet scoring was performed on each sample separately using Scrublet[69]. Briefly, this method simulates synthetic doublets by sampling pairs of cells and then scores each cell by its similarity to the simulated doublets. For each sample, count matrices for RNA and AbSeq data were concatenated and used as the input. The method was modified to exclude known multiplet-tagged cells from the doublet simulation while retaining them for scoring. The doublet score was used to help identify clusters of doublets in subsequent quality control stages.

## Normalisation, integration, dimensionality reduction and clustering

Data normalisation was performed using Seurat[70]. The RNA count matrices were normalised by log normalisation. Specifically, the RNA

counts for each cell were (i) divided by the total counts for that cell, (ii) multiplied by a scale factor of 10,000, and (iii) natural-log transformed after adding one pseudo count. The AbSeq counts for each cell were considered compositional and normalised by a centred log ratio transformation[71]. Normalised RNA and AbSeq expression matrices were then centred and scaled. Specifically, each feature was linearly regressed against selected latent variables including the number of RNA and AbSeq features, and the donor of origin, where appliable. The resulting residuals were mean-centred and divided by their standard deviations.

Integration of the scaled AbSeq expression matrices from different participants was performed using a combination of canonical correlation analysis (CCA) and identification of mutual nearest neighbours (MNNs), implemented in Seurat[70]. After integration was performed, the updated AbSeq expression matrices produced by the integration algorithm were used in subsequent dimensionality reduction and clustering.

Dimensionality reduction and clustering were performed using Seurat[70]. Specifically, principal components (PCs) were calculated separately for each assay using the top variable features of the scaled or integrated expression matrices, and a weighted nearest neighbour (WNN) graph as well as a weighted shared nearest neighbour (WSNN) graph were generated on the ten top PCs from each assay, unless otherwise specified. We used $k = 30$ nearest neighbours for each cell. Uniform Manifold Approximation and Projection (UMAP)[72] embeddings were then calculated based on the WNN graph. The Louvain algorithm[73] was applied to identify clusters of cells based on a WSNN graph, at the selected resolution level. Clusters with fewer than ten cells were removed. The resulting clusters were manually annotated based on marker genes and sorting gate information to elucidate cell types and label suspicious clusters.

### Data quality control

The BD Rhapsody single-cell sequencing data generated in this study was processed through a pipeline consisting of multiple quality control steps as described below. Methodological details of each step are specified in previous sections. The single-cell sequencing data of each sample (i.e. the collection of stimulated and unstimulated cells for a single participant) were pre-processed to obtain two annotated count matrices (RNA and AbSeq). We performed data normalisation, dimensionality reduction and clustering for each sample. Specifically, 30 RNA principal components (PCs) and 30 AbSeq PCs were used to generate the WNN and WSNN graphs, and clustering was performed at resolution 0.5.

Within each sample, cells meeting one of the following criteria were removed: (1) cells labelled as "multiplet" or in clusters with high doublet scores; (2) cells labelled as "tag N/A", except those in clusters annotated as CD56[br] NK cells; (3) cells in clusters with very low RNA counts and very low or very high AbSeq counts; (4) cells in clusters annotated as monocyte or B cell contamination; (5) cells labelled as CD4⁻CD8⁻ double negative (DN) T cells.

The filtered cells from each participant were merged according to the stimulation status into two datasets (unstimulated cells and in vitro stimulated cells). Both stimulated and unstimulated cells were then processed in the same way unless otherwise specified. We performed data normalisation, dimensionality reduction and clustering for each merged dataset. Specifically, clustering was performed at resolution 0.7 (unstimulated cells) or 1.5 (stimulated cells). Doublet clusters were manually annotated and removed. Data normalisation, dimensionality reduction and clustering were performed again. Specifically, clustering was performed at resolution 0.5.

Cells in each filtered dataset were subsequently partitioned into four major groups (CD4⁺ Treg, CD4⁺ Tconv, CD8+ T and CD56⁺ NK cells) based on cluster annotations and sorting gates of origin. In addition, unstimulated cells in clusters marked by genes related to cell cycle were grouped as cycling cells. We chose to partition cycling cells into one separate group based on the following considerations: (i) cycling cells were relatively rare in numbers (0.72% of total unstimulated cells); (ii) cycling cells clusters had well-defined mRNA markers; (iii) The phenotypical differences between cycling cells and their non-cycling counterparts were potentially larger than the heterogeneity within each major cell populations.

After partitioning, doublet clusters were manually annotated and removed. Data normalisation, dimensionality reduction, integration and clustering were performed in each group of partitioned cells. Specifically, clustering was performed at resolution 0.5. Finally, data normalisation, dimensionality reduction, and clustering were performed again.

Owing to the aforementioned technical limitations with AbSeq staining in CD56[br] NK cells, AbSeq data were not used in dimensionality reduction and clustering of NK cells. The following resolutions were used for the final clustering: unstimulated Tregs, 0.6; unstimulated Tconvs, 0.5; unstimulated CD8, 0.5; unstimulated NK, 0.3; unstimulated cycling cells, 0.5; stimulated Tregs, 0.5; stimulated Tconvs, 0.4; stimulated CD8: 0.5; stimulated NK: 0.5. The resulting partitioned datasets were used for subsequent analyses. We note that one PBMC aliquot (corresponding to the Day 55 visit of Participant 8) yielded very low cell numbers ($n = 603$ unstimulated and $n = 67$ stimulated cells) and displayed a compromised FACS staining profile. These cells were included in quality control and clustering procedures but excluded from downstream analyses. All other 38 samples corresponding to 38 visits yielded the expected cell frequencies and numbers.

### Single-cell proliferation scores

Single-cell proliferation scores were calculated using a previously described approach[74], which is based on the average normalised expression levels of a pre-selected set of 11 mRNA features related to cell cycle (*AURKB*, *HMGB2*, *HMMR*, *MCM4*, *MKI67*, *PCLAF*, *PCNA*, *TK1*, *TOP2A*, *TYMS*, and *UBE2C*), subtracted by the aggregated expression of a control set of 50 randomly selected mRNA features. The proliferation scores were used to identify two cycling clusters from the 15 functional T and NK cell subsets identified in blood from the initial clustering (Supplementary Fig. 3a). We noted that the distinct co-expression of these cell-cycle genes present in our targeted transcriptional panel superseded more subtle functional differences and aggregated all cycling T and NK cells into two respective clusters, regardless of their original sorting gate. Given this lack of functional differentiation and to avoid the potential overwhelming effect of these cell-cycle genes on the resulting clustering visualisation, we opted to separate these cycling T and NK cells from the rest of the cells included in subsequent clustering steps to assess IL-2-induced changes in their relative frequency in blood.

### Cluster annotation

Functional annotation of the resulting clusters was performed manually based on the cluster-specific differential expression of the mRNA and protein markers (summarised in Supplementary Data 3). Sample tagging information, corresponding to sorting gate information, was used to sub-cluster the five main immune populations (CD4⁺ CD127[low]CD25[hi] Treg; CD4⁺ CD25[−/low] Tconv; CD8⁺ T; CD56[br] NK and CD56[dim] NK cells). The use of a bespoke targeted single-cell approach combined with a cell-subset enrichment strategy provided a high-resolution map of the heterogeneity of these cell populations – and particularly of the rare circulating CD4⁺ Treg and CD56[br] NK cell subsets.

We note that a consequence of the cell-subset enrichment strategy used in this study is that the frequency of the T and NK cell subsets described in this dataset does not directly represent their frequency in whole blood. The aim of this study was not to identify changes in the global composition of PBMCs, but to specifically

focus on the T and NK cell subsets. The pre-purification of the Treg and CD56[br] NK subsets was central to the resolution of this approach and provided greater power to dissect their functional heterogeneity and investigate how their relative composition was modified by LD-IL-2 immunotherapy

### Functional annotation of CD4+ CD127[low]CD25[hi] T cell clusters

Owing to our cell-subset enrichment strategy, we obtained a high-resolution map depicting the heterogeneity of CD127[low]CD25[hi] T cells in blood, with 13 distinct clusters identified in unstimulated cells (Supplementary Fig. 4a) and 12 in in vitro stimulated cells (Supplementary Fig. 4b). Overall, we found consistency in the functional annotation of the identified subsets in stimulated and unstimulated cells, with a clear demarcation of the naïve and memory subsets.

Expression of the canonical Treg transcription factors *FOXP3* and *IKZF2* (encoding HELIOS) was used to identify three distinct groups of clusters, corresponding to different populations of CD127[low]CD25[hi] T cells: (i) CD45RA+ FOXP3+HELIOS+ clusters (depicted in green) corresponding to thymic-derived naïve Tregs; (ii) CD45RA− FOXP3+HELIOS+ clusters (depicted in blue) corresponding to memory Tregs with suppressive function; and (iii) CD45RA− CD25+ FOXP3−HELIOS− clusters (depicted in orange) corresponding to a small subset of Teffs expressing CD25 likely due to recent activation (Supplementary Fig. 5). Additional clusters showing variable expression of FOXP3 but lack of HELIOS expression (FOXP3+HELIOS− clusters) were depicted in black, and represent a more heterogeneous group of Treg subsets, containing a mix of thymic-derived and induced Treg subsets.

As expected, most of the cells within the CD127[low]CD25[hi] Treg gate showed concomitant expression of FOXP3 and HELIOS, which is consistent with their largely demethylated *FOXP3* Treg-specific demethylated region (TSDR)[34], the hallmark of bona fide thymic Tregs. Furthermore, we were able to identify a trajectory of thymic Treg differentiation, originating from the naïve FOXP3+HELIOS+ Tregs (cluster 0). Memory FOXP3+HELIOS+ Tregs displayed more heterogeneity and included four distinct subsets in unstimulated cells, including a cluster of CD80+ Tregs characterised by elevated expression of tissue-homing receptors (cluster 7) and a cluster of effector Tregs (cluster 1) typically marked by the expression of HLA class II and other effector Treg markers such as CD39 and GITR (Supplementary Fig. 4a). We have also identified a subset of CD45RA+ naïve cells (cluster 2) showing much lower expression of FOXP3 and HELIOS compared to the classical naïve Treg cluster. This likely corresponds to a subset of homeostatically-expanded CD45RA+ T cells upregulating CD25 that we have previously characterised[75]. Consistent with this non-Treg profile, these CD25+ naïve T cells upregulated the expression of IL-2 upon in vitro stimulation (Supplementary Fig. 5f). The expression of IL-2 in stimulated cells was also a hallmark of the CD25+ FOXP3−HELIOS− Teff clusters and allowed us to further refine their functional heterogeneity. Moreover, the relative frequency of CD45RA− FOXP3−HELIOS− cells identified in this single-cell dataset is consistent with the frequency of cells identified by FACS as cytokine-producing FOXP3−HELIOS− cells in the CD127[low]CD25[hi] Treg gate that we have previously shown to contain a methylated TSDR[34], further indicating that they correspond to a small proportion of activated CD25+ FOXP3−HELIOS− Teff cells captured in the CD127[low]CD25[hi] Treg gate.

### Functional annotation of CD56+ NK cell clusters

In contrast to T cells, we found much lower heterogeneity in NK cells (Fig. 4d and Supplementary Fig. 14). In addition, the technical issue affecting the efficiency of the AbSeq staining on CD56[br] NK cells, precluded the use of protein data for the clustering and annotation of the CD56+ NK cells, which also contributed to a lower resolution of the clusters. Nevertheless, our cell-enrichment strategy allowed us to generate a comprehensive single-cell map of NK cell heterogeneity,

most notably of the rare, IL-2-sensitive CD56[br] NK population. We observed differentiation of CD56[br] NK cells from a rare progenitor subset marked by the expression of genes associated with haematopoiesis such as SOX4 and KIT, and characterization of two functional subsets associated with a trajectory of CD56[br] NK cell maturation, which was marked by HLA class II gene expression in the more differentiated cells. In contrast, CD56[dim] NK cells showed a distinct transcriptional profile of two distinct subsets, reflecting a gradient of NK cell activation associated with the acquisition of a cytotoxic profile (Fig. 4d and Supplementary Fig. 14). These findings support a distinct functional differentiation between the CD56[dim] and CD56[br] subsets, which allowed clear identification a fraction of CD56[dim] NK cells sorted in the CD56[br] gate, as well as a fraction of CD56[br] NK cells sorted within the CD56[dim] gate (Fig. 4e).

### Differential abundance analysis

To identify specific cell frequency changes across different sampling time points, we calculated the frequency of each cell type annotated. A cell type was defined as a group of cells from the same cluster or several similar clusters, after removing cells not sorted from the corresponding gate. The frequency of each cell type for each participant and each time point was calculated as the proportion of that cell type within all cells from the same participant, time point and sorting gate of origin. For each cell type, a two-sided Wilcoxon signed-rank test was performed to compare the frequencies at Day 0 with the frequencies at Day 27 or Day 55 in the 13 participants selected for single-cell analysis. Benjamini-Hochberg FDR correction was applied after pooling all resulting $P$ values. For each cell type and each participant, the $\log_2$ fold change values between corresponding frequency values were calculated and visualised. We note that cells were compared only if they were from the same sorting gate, as cells from different sorting gates were pooled in designated proportions that are not biologically relevant.

### Generation and normalisation of pseudo-bulk expression data

Unstimulated and stimulated cells were separately partitioned into five groups (Treg, Tconv, CD8, CD56[br] and CD56[dim]), after removing cells not sorted from the corresponding gate. In each group, cells from the same participant, time point and stimulation status were aggregated into one pseudo-bulk sample by summarising raw counts of each mRNA or AbSeq feature.

### Differential expression analysis

Differential expression analysis was performed separately for RNA and AbSeq data from raw pseudo-bulk counts using DESeq2[76]. Specifically, the likelihood ratio test was used, with a full model including time points and the participants as independent variables, and a reduced model including only participants as the independent variable. The apeglm method[77] was applied to shrink the resulting fold change values, and the Benjamini-Hochberg FDR correction was applied after pooling all resulting $P$ values (Supplementary Data 4). Genes with either (i) absolute log2 fold change values equal to or greater than 0.4 and FDR-adjusted $P$ values less than 0.01, or (ii) FDR-adjusted $P$ values less than $1 \times 10^{-10}$, were selected as significantly differentially expressed genes, unless otherwise noted. Participant-specific $\log_2$ fold change values were calculated directly from the normalised pseudo-bulk expression data.

### Day 55 signature scores

Forty-one genes that were significantly upregulated or downregulated in at least one unstimulated cell population comparing Day 55 with Day 0 were defined as Day 55 signature genes, from which the Day 55 signature scores were calculated. Within each dataset, the Day 55 signature score was defined for each pseudo-bulk sample as the sum of z-scores of normalised expression levels of upregulated signature genes, subtracted by that of downregulated signature genes. For each participant

and cell type, the Δ signature scores were defined as the signature score differences between Day 27 or Day 55 and Day 0, as indicated.

## Pseudo-time trajectory analysis

Pseudo-time trajectory analysis was performed using Slingshot[78] and tradeSeq[79]. First, 10-dimensional UMAP embeddings were generated for stimulated CD4[+] T cells based on RNA and AbSeq data, as described above. Second, the Slingshot algorithm was applied to UMAP data and the manually annotated cluster labels, with naïve cells selected as the starting point, and other parameters set to default. Finally, the resulting pseudo-time data were used as the input for the tradeSeq algorithm to model the expression levels of each gene along each trajectory.

## Reporting summary

Further information on research design is available in the Nature Research Reporting Summary linked to this article.

## Data availability

The single-cell sequencing data generated in this study have been deposited in Gene Expression Omnibus (GEO) under accession code GSE201197. Source data are provided with this paper.

## Code availability

The custom code used in the study for data analyses and visualisation has been deposited in FigShare (https://doi.org/10.6084/m9.figshare.21395214).

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

## Acknowledgements

We thank all participants in DILT1D and DILfrequency for their generous contribution to this study. The authors also acknowledge the support of the NIHR Cambridge Biomedical Research Centre and the Cambridge Clinical Trial Unit for trial coordination; the NIHR/Wellcome Trust Clinical Research Facility and Addenbrooke's Centre for Clinical Investigation for clinical facilities; the Department of Clinical Immunology, Addenbrooke's Hospital; the JDRF/Wellcome Diabetes and Inflammation laboratory sample processing team (led by Helen Stevens) and information technology and administration team (led by Judy Brown) and data teams at the Cambridge Institute for Medical Research. We thank Shannah Donhou, Sarune Kacinskaite and Heather McMurray, University of Oxford for sample collection and preparation, and members of the Diabetes and Inflammation Laboratory for critical discussion. This work was supported by the Sir Jules Thorn Trust (13/JTA (OCT2013/DR/1044)), the JDRF (1-SRA-2019-657-A-N), and the NIHR Cambridge Biomedical Research Centre. The Diabetes and Inflammation Laboratory was supported by a strategic award from the Wellcome (107212/A/15/Z) and the JDRF (4-SRA-2017-473-A-A). J-Y.Z. was supported by the China Scholarship Council-University of Oxford Scholarship. The research was supported by the Wellcome Trust Core Award Grant Number 203141/Z/16/Z with additional support from the NIHR Oxford BRC. The University of Cambridge has received salary support for M.L.E. through the National Health Service in the East of England through the Clinical Academic Reserve. The views expressed are those of the author(s) and not necessarily those of the NHS, the NIHR or the Department of Health.

## Author contributions

L.S.W., J.A.T. and R.C.F. conceptualised this work and designed the experiments. M.L., L.G. and R.C.F. performed the scRNA-seq experiments. J-Y.Z. led the analysis of the scRNA-seq data with help from F.H. and D.T. M.M. and J.H.M.Y. performed the in vitro suppression experiments and analysed the data under the supervision of T.I.M.T. M.L.P. and R.C.F. performed FACS experiments and analysed the data. J.K., F.W-L. and M.L.E. supervised the DILfrequency study. T.I.M.T. provided conceptual advice on the experiments. J-Y.Z., L.S.W, J.A.T. and R.C.F. wrote the manuscript with input from all authors.

## Competing interests

F.W-L. is employed by Vertex Pharmaceuticals; J.A.T. is a member of the GSK Human Genetics Advisory Board. The remaining authors declare no competing interests.
