## [Peer Review File · Nature Communications]

Low-dose IL-2 reduces IL-21⁺ T cell numbers and induces anti-inflammatory gene expression in type 1 diabetesREVIEWER COMMENTS

Reviewer #1 (Remarks to the Author)

This manuscript conducted a single-cell analysis of the immune responses to interval low-dose IL2 treatment based on 18 participants. While it is interesting to illustrate the effects of IL2 treatment, the experimental design is limited by customized gene panels and selected single cells, which might introduce artificial biases prohibiting the assessment of the IL2 treatment. Although COVID-19 data were evaluated, the data were published and did not relate to IL2 treatment.

Reviewer #2 (Remarks to the Author):

This work by Zhang and colleagues intends to broaden current knowledge about the mode of action of a low-dose IL-2 regimen in patients with type-1 diabetes (T1D) by using classical flow cytometry and a single cell multiomics approach. The rationale behind this work is sound and the topic itself is of high interest. In principle, most applied methods appear to be adequately performed. The main findings are that this low-dose IL-2 regimen selectively expands CD127^{lo}CD25^{hi} and FoxP3⁺Helios⁺ Treg cells as well as CD56^{br} NK cells, decreases subsets of T follicular helper cells which produce IL-21, and induces a sustained anti-inflammatory signature which could be detected even 4 weeks after the end of the IL-2 treatment.

However, although relevant and of specific interest in the context of T1D, many of the findings described here have already been reported in other diseases such as SLE and thus are not entirely novel from a more global perspective. In addition some of the conclusions seem not justified based on the provided data. In particular, I do not understand the link between LD IL-2/T1D and COVID-19 infection, as both diseases have a completely different pathophysiology and etiology. Thus, the creation of such inverse relationship appears artificial and is rather distracting from the rest of findings in T1D. I would thus suggest removing the COVID-19 data from the manuscript as it's rather a story on its own (even without these data the manuscript is still quite extensive).

Specific comments:

Methodology

1. FACS gating and staining strategy: Why was FoxP3 staining in combination with CD127 not used for the definition of the Treg population, which now for a long time is considered to be the state-of-the-art staining for the reliable identification of Treg by flow cytometry and for the discrimination between Treg and conventional CD4⁺ T cells. The inclusion of FoxP3 staining is essential from my point of view as a large proportion of genuine FoxP3⁺ Treg does not express or expresses only low levels of CD25, which are completely excluded from the analysis if only CD25^{hi}CD127^{lo} cells are gated, and activated CD4⁺ effector T cells can express high levels of CD25, which was also taken into account by the authors in the multiomic analyses. In light of this and of the presented data derived from flow cytometry, the flow cytometric analyses appear a bit rudimentary from my point of view, compared to those from other published clinical trials (see also Grasshoff et al. for detailed summary of findings from trials). At least this methodical limitation should be mentioned and explained in the discussion. Why was the proliferation of Treg and other cells that can respond to IL-2 not assessed by including the commonly used proliferation marker Ki67 in the FACS analyses?

2. Identification of Tfh cell subsets: Although I appreciate the efforts of the multiomics approach, I wonder a bit why the authors did not use classical flow cytometry here which is capable to reliably analyse changes in frequencies and numbers of Tfh subsets at a single cell level, instead of going the complex way of multiomics, which might also be susceptible to misinterpretations, because of the very low frequencies of these subsets in their presorted populations and the high variability of obtained data.

3. Patient selection: Why have only 18 participants from the trial been selected and only 13 for the multiomics approach? How many patients in total have been treated in the trial? With regard to the assessment of post treatment effects the inclusion of an intermediate time point between day 27 and day 55 would have been valuable.

Findings

1. At the level of Treg and NK cells including the finding that low dose IL-2 expands thymic-derived Helios+ Treg and the CD56br NK cell subset, this study only confirms what is already known in T1D and other autoimmune disease treated with LD IL-2, and thus does not add much novel aspects to this (eg von Spee-Mayer Ann Rheum Dis 2016, Humrich Lancet Rheumatol 2019, He Ann Rheum Dis 2019). Apart from this, analysis of Helios expression and of other Treg markers by classical multi-color flow cytometry would certainly have produced more accurate data on frequencies and numbers of such cell subsets than by using single-cell multiomics.
2. The finding that LD IL-2 is capable to decrease Tfh subsets is also not entirely new, in contrast to the claim of the authors, as this was previously described in SLE patients treated with LD IL-2 therapy (He Nat Med 2016, Humrich Lancet Rheumatol 2019). Given the rather marginal reductions (Fig. 3f) of IL-21 producing CD4+ T cell subsets and the high variability, which was found by the single cell multiomics approach, I think it would be necessary to confirm this finding by other approaches, eg is there also a reduction of IL-21 levels in plasma (if measurable) or of IL-21 producing CD4+ T cells assessed by flow cytometry in PMA/ionomycin stimulated cells.
3. The findings on CD8+ T cells are of interest and appear also novel. I would discuss these findings a bit more in detail, also in the context of T1D.
4. From my point of view, the most interesting and also novel finding of this study was the sustained anti-inflammatory signature present 4 weeks after the treatment including the up-regulation of CISH in Treg and other cells and the downregulation of TNF induced genes. However, as these findings have not been further addressed in detail by other approaches or by correlating them with clinical responses (why are those not shown or mentioned?), they remain only of a descriptive nature. I also miss in the discussion the relation of these findings with T1D pathophysiology. Overall, I feel that the use of the rather complex single cell multiomics approach in this setting did only provide little new and also rather preliminary information on the mode of action of IL-2, which needs to be confirmed somehow by other approaches.
5. Dose responses: I think it is not justified to speak about dose dependent effects, considering that there are often only 2 data points (eg Fig 6d, Fig. S9e) for the lowest and highest dose in the analyses and that there is a high inter-individual variability in the multiomics data.

Conclusions/Discussion

Many of the conclusions in the discussion seem to be speculative, eg that this 3-day interval dosing regimen is better than daily or cyclic regimens. This does not fit to what was found in SLE patients by the use of a repetitive 5-day LD IL-2 regimen which was very effective in selectively expanding Treg without any hints for desensitization (Humrich Lancet Rheumatol 2019). To address this issue a direct comparison between the regimens would be required. In particular, the COVID-19 data interpretation is highly speculative and should be removed as suggested above. From my personal view I would not suggest to give IL-2 to patients with acute and severe hyperinflammation as observed in COVID-19 infections, as this could be like putting oil into fire.

Terminology

1. The terminus "effector" T cell is a bit misleading in this context, as those non-Treg CD4+ T cells also contain many naïve T cells. I would suggest using the term conventional T cell (Tcon or Tconv) instead.
2. I would not use the term "depletion" for describing the reduction of cell subsets, as this suggests that these cells are actively removed or killed, but most likely there is just less generation or differentiation of these subsets.

Reviewer #3 (Remarks to the Author):

In the proposed paper the authors study the effects of low dose IL-2 in T and NK cells. They apply a targeted single-cell approach. The paper is well written and the study is very detailed. Furthermore authors are very precise in describing the few technical limitations and clearly show how they solved the related issues. I have just a few questions/comments:

- In the UMAPS depicting the clustering of unstimulated and stimulated cells (Supplementary Figure 2) is there any population/cluster appearing after treatment (or any time/dose-specific population?)
- Both in the case of Tfh and Treg have the authors tried to perform a trajectory analysis confirming the differentiation of the populations?
- Why the authors did not decide to regress out cell cycle genes from the analysis instead of excluding the clusters of cycling cells in a later stage of the analysis ?
- Supplementary figure 2 legend sentence "Untagged cells in b (labelled as "Tag N/A" correspond to ..." should be "Untagged cells in a" ...)
- In the printed version of the paper (A4) Labels in Supplementary figure 5 (boxplots) are not readable

NCOMMS-22-13925-T

Point-by-point reply to reviewers' comments

Reviewer #1

1. This manuscript conducted a single-cell analysis of the immune responses to interval low-dose IL2 treatment based on 18 participants. While it is interesting to illustrate the effects of IL2 treatment, the experimental design is limited by customized gene panels and selected single cells, which might introduce artificial biases prohibiting the assessment of the IL2 treatment.

Although we acknowledge the different applications of whole transcriptome (WT) and targeted scRNA-seq methods, we do not believe that the targeted approach can lead to spurious results that preclude the investigation of the IL-2 treatment in this study. Targeted scRNA-seq methods (based both on the BD Rhapsody and other competitor systems) are now widely used and have shown to provide increased sensitivity compared to WT methods, especially for lowly expressed genes. As an illustrative example, we have empirically compared the sensitivity of both methods to quantify the expression of the transcription factor FOXP3, showing a significantly increased detection rate for the targeted assay (68% vs 23% in cells from the identified Treg cluster; see Trzupsek D et al. *Genome Med* 2020). Furthermore, the protein quantification feature - using the AbSeq technology - has also been shown to precisely recapitulate flow cytometry data (see Mair F et al. *Cell Reports* 2020 for a detailed technical comparison), which further increases the sensitivity of this single-cell multiomics approach. This is particularly relevant in the context of this study, as it provides increased power to dissect the functional heterogeneity of T and NK cells, which have previously been challenging to dissect using traditional WT methods.

In this study we set out to extend previous findings from flow cytometry-based studies of low-dose IL-2 (LD-IL-2) in both type 1 diabetes (T1D) and other autoimmune diseases. Learning from those previous studies, we wanted to focus on specific target populations, namely CD3⁺ T cells and CD56⁺ NK cells, which have been shown to be modulated in frequency and numbers by LD-IL-2 immunotherapy. Such cell subset enrichment strategy has long been employed in immunology in FACS-based experiments to increase the power of the study. Our

cell subset enrichment strategy was specifically designed in order not to exclude any cell from the broad CD3⁺ T cell and CD56⁺ NK cell populations. Although CD56^{br} and CD4⁺ Treg subsets were enriched in our study, we were able to trace the compositional changes of each subset after IL-2 treatment through cellular barcodes specific to each sorting gate.

The number of cells profiled in this study, especially from the CD56^{br} NK and CD4⁺ Treg populations, could not have been achieved using a WT approach. Many of the cellular alterations identified by our multiomics approach replicated previously identified changes by FACS in other LD-IL-2 studies, which provides a technical validation of our assay. Furthermore, we have replicated several of our key immune populations using a FACS-based approach, showing a very good consistency between the two methodologies. Details can be found in our replies to Reviewer #2 below. We believe these new data will help alleviate the concerns that the targeted single-cell multiomics approach employed here could compromise the technical integrity of our study due to erroneous or spurious results.

2. Although COVID-19 data were evaluated, the data were published and did not relate to IL2 treatment.

The main objective of presenting the COVID-19 data was to support the existence of the identified IL-2-induced long-lived transcriptional signature by demonstrating that a different immune perturbation (a pro-inflammatory viral infection in the example we found) caused a highly, albeit negatively, correlated signature. As the reviewer correctly points out, our observation that the expression of the same set of target genes is reversed in COVID-19 patients does not directly prove that the mechanism is driven uniquely by IL-2. We believe that the widespread gene expression changes observed in our study likely reflects the sustained cellular alterations during the treatment period, leading to a modification of the homeostatic regulation of the immune system. We understand this limitation of our study and more clearly highlight it in our revised discussion.

In order to further strengthen our findings, we now also show additional data from a large longitudinal cohort of convalescent COVID-19 patients (INCOV cohort – see Su Y et al. *Cell* 2022) that was recently published. These additional data not only replicate our original finding in the COMBAT cohort, but also provide a much better understanding of the dynamics of the

induction of the transcriptional changes after the onset of the clinical symptoms. In particular, the longitudinal cohort of convalescent patients reveals a rapid induction of the transcriptional response within the first 2-3 weeks after the onset of symptoms, followed by a gradual but slow recovery towards homeostatic baseline levels over the next few months.

The COMBAT study did find Component 187, which we show is highly consistent with our reported IL-2-induced day 55 gene expression signature), but it is not the case for the INCOV cohort, where there is no report of such a long-standing transcriptional signature. To better discriminate between the LD-IL-2 and COVID-19 results, we have revised the main text figure (**Fig. 6**) to focus our primary analysis exclusively on our identified Day 55 gene expression signature (i.e. the 41 differentially expressed genes in response to LD-IL-2 at Day 55). We derive a signature score based on the genes identified exclusively from this signature and compare the same set of genes in the COMBAT and INCOV datasets (now shown in the new **Fig. 7**). We believe that this approach illustrates more clearly that the same genes are induced (in opposite direction) in COVID-19 patients, therefore supporting the induction of anti-inflammatory gene expression during and after LD-IL-2 immunotherapy. Yes, this is available published data, but the particular analyses, emphasis and comparison we present are novel.

Although the mechanism underlying the induction of the long-lived transcriptional signature is still incompletely understood, we believe that the strength and consistency of the reverse association is relevant and of interest for a wider audience - particularly in the context of the increasing number of patients affected by long-term sequelae of COVID-19. We have substantially revised the Discussion section to explain the hypothesised link between the alteration of gene expression in the context of IL-2 treatment and COVID-19 more clearly.

Reviewer #2:

General Comment:

1. [...] However, although relevant and of specific interest in the context of T1D, many of the findings described here have already been reported in other diseases such as SLE and thus are not entirely novel from a more global perspective.

We have revised the manuscript to refrain from claiming novelty in certain of our findings, and referred to the original publications in SLE. We thank the reviewer for pointing out that a modulation of CD4⁺ T_{FH} cells has been reported by FACS in SLE patients following LD-IL-2 treatment. This was an important omission, which reflects some key disease-specific differences in the response to LD-IL-2. In other revisions we have made clear which particular results and insights are novel (IL-21 production inhibition at the cell-specific level; and re-analyses of two COVID-19 datasets) and which confirm previous findings. For example, we now show that the reduction of CD4⁺ T_{FH} cells after IL-2 treatment is driven mainly by a reduction of the more differentiated IL-21-producing T cells. This increased power to identify more subtle relative compositional changes is likely responsible for the ability to identify a similar modulation of IL-21⁺ T cells in T1D in response to our IL-2 treatment, which had not been previously reported by traditional flow cytometric assessment.

2. In addition some of the conclusions seem not justified based on the provided data. In particular, I do not understand the link between LD IL-2/T1D and COVID-19 infection, as both diseases have a completely different pathophysiology and etiology. Thus, the creation of such inverse relationship appears artificial and is rather distracting from the rest of findings in T1D. I would thus suggest removing the COVID-19 data from the manuscript as it's rather a story on its own (even without these data the manuscript is still quite extensive).

We never expected a long-lived effect of our iLD-IL-2 dosing regimen and since we did not have sufficient numbers of remaining PBMC samples for additional, well-powered single-cell analyses to replicate the finding in the context of IL-2 therapy, we were relieved and excited to see exactly the same genes being altered in the COMBAT study. And then the replication in the INCOV study really does support the existence of differential expression of these genes in every cell type analysed. The cellular and phenotypic consequences of this transcriptional regulation remain to be determined in future investigations.

We have substantially edited the manuscript in a number of ways to clarify the relevance of the COVID-19 findings and to better explain the potential mechanism underlying the induction of the expression of the same genes (in opposite direction) in these two very different contexts. Namely:

1/ We have revised our figures to clearly separate the finding of the long-standing gene expression signature in the LD-IL-2 and COVID-19 settings. We have done this by removing the COVID-19 data from current **Fig. 6** and creating a new figure (**Fig. 7**) containing only COVID-19 data. This allows to focus **Fig. 6** on the identification of a long-lasting anti-inflammatory signature following IL-2 treatment, which we agree is the more novel finding from this study and a wider contribution to the field of IL-2 immunotherapy.

2/ To further substantiate the finding from the COMBAT cohort, we have now replicated the same result in a different longitudinal cohort of COVID-19 patients that has been recently published (INCOV cohort – see Su Y et al. *Cell* 2022). Importantly, this represents the first large single-cell multiomics dataset of convalescent patients and demonstrates not only the induction of same LD-IL-2-altered genes in reversed direction – in all immune cell populations – but also reveals the longevity of the gene expression. These data are now summarised in the new **Fig. 7** along with the COMBAT data to better separate the findings pertaining to COVID-19. The COVID-19 analyses are essential for the IL-2 finding – they convincingly support that long-lived alteration of these immune response genes does occur in our LD-IL-2 regimen and that it has a wide immunological relevance since IL-2 treatment and virus infection are two very different physiological events. In addition, the replication evidence provided by the second COVID-19 cohort provides firm support for the gene expression reported in COMBAT.

3/ As mentioned on the reply to R1 Q#2, we have revised the analysis of the COVID-19 data to focus specifically on the set of genes induced by LD-IL-2 treatment (i.e., the 41 differentially expressed genes in response to LD-IL-2 at Day 55; shown in **Fig. 6**). This better demonstrates the strong link between the induction of the same set of genes (in opposite direction) in these two settings.

4/ We now provide a formal analysis to show that the overlap between the 41 differentially expressed genes that compose the Day 55 signature is restricted to the cluster of multiparametric signatures associated with the COMBAT Component 187 (which we show is strongly negatively correlated with the IL-2-induced Day 55 signature score). There is no evidence for overlap with any additional COMBAT signature, thereby confirming the very distinct pathophysiological mechanism between COVID-19 and LD-IL-2 immunotherapy.

5/ Given the strong pro-inflammatory response in COVID-19 patients, the reversed expression of the same genes in this context therefore provides strong support for the anti-inflammatory nature of the gene expression changes induced by LD-IL-2, which is strengthened by the replication of the results in the INCOV dataset. Although we acknowledge the limitation that we cannot elucidate the exact mechanism driving this opposite response in the two contexts, we believe that the results and the longevity of the transcriptional changes are intriguing and warrant further investigation; particularly considering the growing concern of the long-term complications in COVID-19 patients. We have therefore revised the discussion to more clearly explain how we hypothesise that the LD-IL-2 and COVID-19 findings could be related and to better enunciate the limitations of the current study.

6/ Finally, we have removed the reference to COVID-19 from the title. We agree that the previous title may have inaccurately hinted that the same gene expression signature was modulated directly by IL-2 in both contexts.

We believe these changes will provide more clarity on the link between the LD-IL-2 and COVID-19 results to the readers.

Specific Comments:

Methodology 1. FACS gating and staining strategy: Why was FoxP3 staining in combination with CD127 not used for the definition of the Treg population, which now for a long time is considered to be the state-of-the-art staining for the reliably identification of Treg by flow cytometry and for the discrimination between Treg and conventional CD4+ T cells. The inclusion of FoxP3 staining is essential from my point of view as a large proportion of genuine FoxP3+ Treg does not express or expresses only low levels of CD25, which are completely excluded from the analysis if only CD25^{hi}CD127^{lo} cells are gated, and activated CD4+ effector T cells can express high levels of CD25, which was also taken into account by the authors in

the multiomic analyses. In light of this and of the presented data derived from flow cytometry, the flow cytometric analyses appear a bit rudimentary from my point of view, compared to those from other published clinical trials (see also Grasshoff et al. for detailed summary of findings from trials). At least this methodical limitation should be mentioned and explained in the discussion.

The use of FOXP3 immunostaining is unfortunately not possible with current single-cell transcriptomics platforms. The cell permeabilization and fixation step required for the intracellular staining affects the integrity of the mRNA, and consequently precludes its use for these assays. For this reason, we resorted to the use of the conventional surface markers CD127 and CD25 to define the CD127^{low}CD25^{hi} Treg gate. Although this sorting strategy unquestionably leads to increased heterogeneity within the CD127^{low}CD25^{hi} subset, we note that such heterogeneity is beneficial for this assay, as the underlying distinct functional subsets can be discriminated by single-cell multiomics, allowing us to precisely quantify the relative compositional changes in the CD127^{low}CD25^{hi} population after LD-IL-2 treatment.

Because FOXP3 is such an important marker to define the different Treg (and non-Treg) subsets, we were particularly interested in assessing the sensitivity of this multiomics method to quantify its transcriptional levels. In a previous publication using this platform (see Trzupke D et al. *Genome Med* 2020), we have shown a ~68% detection rate of *FOXP3* mRNA in cells sorted from the CD127^{low}CD25^{hi} gate, which is close to the frequency of FOXP3⁺ cells identified by FACS. We therefore believe that the use of *FOXP3* mRNA in our single-cell multiomics system is an adequate surrogate for the protein expression and, in combination with the additional mRNA and protein (AbSeq) markers available, allow a robust discrimination of the FOXP3⁺ Treg and FOXP3⁻ CD25⁺ Tconv subsets within the CD127^{low}CD25^{hi} population.

With regards the CD25^{-/low} FOXP3⁺ cells, we have a long-standing interest in this population, given its increased frequency in autoimmune diseases, including T1D. However, in blood from T1D patients, the frequency of CD25^{-/low} FOXP3⁺ cells is extremely low (usually <1-2% of total CD4⁺ T cells). However, given our cell sorting strategy, these CD25^{-/low} FOXP3⁺ cells were not excluded from the analysis and captured in the CD25^{-/low} CD4⁺ Tconv cell gate, containing the vast majority of CD4⁺ T cells. Owing to their very low number, we were not able to identify a specific cluster representing the CD25^{-/low} FOXP3⁺ Tregs. We have previously shown that CD25^{-/low} FOXP3⁺ Tregs represent a bona-fide Treg population, with demethylated FOXP3

TSDR (see Ferreira RC et al. *J Autoimmun* 2017), and cannot be distinctly discriminated from conventional CD25^{hi} FOXP3⁺ Tregs using single-cell transcriptomics – likely due to their very similar transcriptomics profile (see Trzupek D et al. *Wellcome Open Res* 2021).

To better identify the CD25^{-/low} FOXP3⁺ cells and whether they are modulated by LD-IL-2, we have refined our analysis to identify all cells with detectable expression of FOXP3 (≥ 1 UMI) within the cells sorted from the CD25^{-/low} CD4⁺ Tconv gate. Using this approach, we confirm the low frequency of CD25^{-/low} FOXP3⁺ cells and the lack of evidence for their modulation by our IL-2 treatment (see **Supplementary Fig. 11a,b**).

Furthermore, to increase the number of assessed CD25^{-/low} FOXP3⁺ cells, we also employed a FACS-based approach to quantify the intracellular expression of the Treg transcription factors FOXP3 and HELIOS in two follow-up replications cohorts: (i) a single-dosing cohort consisting of six participants from our original DILT1D trial (see Todd JA et al. *PLoS Med* 2016) treated with IL-2 doses ranging from 0.6 to 1.5 x 10⁶ IU IL-2/m²; and (ii) a multiple dosing cohort consisting of five selected DILfrequency participants treated with the 3-day dosing regimen where PBMC samples were still available. Patients were profiled at 10 (Days 0, 1, 2, 3, 4, 5, 7, 9, 14, 28) and 4 (Days 0, 3, 27 and 55) timepoints for the DILT1D and DILfrequency cohorts, respectively. These FACS data confirmed that CD25^{-/low} FOXP3⁺ Tregs represent a very small percentage of total CD4⁺ T cells in blood from T1D patients (<2% see new **Supplemental Fig 11d,e**) and showed little variation in response to LD-IL-2 – either as % of CD4⁺ T cells or as the frequency of FOXP3⁺ cells within the CD25^{-/low} Tconv gate - both at the initial and later phases of the treatment period (see **Supplemental Fig 11d-g**).

Together, these results demonstrate the consistency in the characterisation of CD25^{-/low} FOXP3⁺ T cells between the multiomics and FACS-based methods. We now summarise these results in the new **Supplementary Fig. 11**, to better highlight this relevant, but often overlooked, cell population and to show that it is minimally affected by our LD-IL-2 dosing regimen.

Methodology 2. Why was the proliferation of Treg and other cells that can respond to IL-2 not assessed by including the commonly used proliferation marker Ki67 in the FACS analyses?

We thank the reviewer's suggestion and have now assessed proliferation by intracellular staining of Ki-67 in our single and multiple dosing replication cohorts and have added them to the results. We demonstrate that there is a substantial discrepancy between the number of proliferating cells identified by either FACS (based on Ki-67 antibody staining) or single-cell multiomics (based on multiple mRNA markers including *MKI67*). These results are very consistent with previous literature, showing that although Ki-67 protein synthesis is restricted to the S and G2/M phases of the cell cycle, the accumulation and longer half-life of the protein can also lead to the detection of Ki-67⁺ cells during the longer and more heterogeneous G1 phase (see di Rosa F et al. *Front Immunol* 2021). Notably, in another study, when comparing the frequency of actively proliferating cells in peripheral blood by FACS and a more specific DNA-based method, the authors report a very similar discrepancy in numbers as we show in this study i.e. ~37% of activated fraction II Tregs were found to be Ki-67⁺ by FACS, whereas only ~1% was found to be in the S-G2/M phases by the DNA-based method; see Munoz-Ruiz M et al. *J Autoimm* 2017.

These results are now summarised in the new **Supplementary Fig. 3**. We believe these results provide a better description of the role of our LD-IL-2 dosing regimen on the proliferation of the assessed immune subsets and a better discussion of these results in context of the previous LD-IL-2 studies. Importantly, we also discuss the reasons for the discrepancy between FACS (protein) and single-cell multiomics (mRNA) methods in the identification of proliferating cells, which is likely relevant for a wider range of studies.

Methodology 3. Identification of Tfh cell subsets: Although I appreciate the efforts of the multiomics approach, I wonder a bit why the authors did not use classical flow cytometry here which is capable to reliably analyse changes in frequencies and numbers of Tfh subsets at a single cell level, instead of going the complex way of multiomics, which might also be susceptible to misinterpretations, because of the very low frequencies of these subsets in their presorted populations and the high variability of obtained data.

In T1D the phenotype of circulating Tfh cells is not very pronounced – consistent with a precursor Tfh population - and is challenging to accurately describe by flow cytometry. Although there are reports showing an increase of Tfh cells in T1D, these corresponded to very subtle differences and required the induction of IL-21 expression to better delineate

circulating Tfh cells. In the context of low-dose IL-2 treatment, no changes in Tfh cells – as described by flow cytometry – have been previously reported. In contrast, in SLE, the phenotype of Tfh cells is much more pronounced (see He J et al. *Nat Med* 2016), reflecting a heightened contribution of germinal centre reactions and activated B cells to disease pathogenesis.

Given the results reported by He J et al. (*Nat Med* 2016) and Humrich JY et al (*Lancet Rheumatol* 2019) and our results in the present study, we sought to replicate these findings using FACS data previously obtained from whole-blood immunostaining at each visit from all 18 DILfrequency donors treated with the 3-day interval dosing regimen. Consistent with the less differentiated phenotype of circulating Tfh cells in T1D, we did not observe such pronounced expression of PD-1 and reduced expression of CCR7 – we used CD27 as a better marker in cryopreserved cells to differentiate central / effector memory T cells - in the CXCR5+ memory T cells – as defined in He J et al. (*Nat Med* 2016). We therefore defined the Tfh subset as the PD-1⁺ fraction among the CXCR5⁺ memory CD4⁺ Tconvs (see **Reviewer Figure 1a** provided in appendix). In agreement with our multiomics data, we observed a very small decrease of Tfh cells at day 27 (See **Reviewer Figure 1b**). To further confirm these flow results, we also assessed the expression of the Tfh delineating markers CXCR5 and PD-1 on the same panel used to phenotype the intracellular expression of FOXP3 and HELIOS in the additional five DILfrequency patients. These data replicated our findings in the larger cohort and showed a small 6.3% decrease in the frequency of Tfh cells (similarly defined as CXCR5⁺ PD-1⁺ memory Tconvs) at day 27 (see **Reviewer Figure 3c**).

We believe that the main strength of our multiomics approach is to better discriminate the heterogeneity of blood Tfh subsets. Furthermore, our approach of stimulating cells *in vitro* allowed us to assess IL-21-producing cells and show that the specific inhibition by LD-IL-2 is restricted to the more differentiated IL-21-producing cells. Our findings therefore expand on the previous data showing reduced frequency of Tfh cells following LD-IL-2 treatment, demonstrating that the effect is not associated with the circulating precursor Tfh cells or with the T follicular regulatory (Tfr) population (as we clearly demonstrate in the multiomics analysis that none of the decreased Tfh subsets expresses FOXP3), but is rather restricted to the more differentiated (IL-21 producing) Tfh cells, which are more likely to become pathogenic. Furthermore, to better support the hypothesised reduction in the differentiation

of IL-21-producing cells, we have performed a pseudo-time trajectory analysis in *in vitro* stimulated cells, which identified a trajectory of differentiation of IL-21-producing Tfh cells that is specifically inhibited by LD-IL-2 immunotherapy (see new **Supplementary Fig. 8**). Together these data provide compelling evidence for the modulation of differentiated IL-21-producing cells by LD-IL-2 treatment, which is an important mechanism for its therapeutic effect.

Methodology 4. Patient selection: Why have only 18 participants from the trial been selected and only 13 for the multiomics approach? How many patients in total have been treated in the trial? With regard to the assessment of post treatment effects the inclusion of an intermediate time point between day 27 and day 55 would have been valuable.

Patient selection was constrained by both the number of donors treated with the same 3-day dosing interval ($N = 18$) and the design of the single-cell multiomics experiment. Twenty patients involved in the earlier adaptive dosing phase of the study were not included, considering that most of them (i) no longer have PBMC samples available, and (ii) were treated with different, sub-optimal dosing schemes, making direct comparison challenging.

To increase the power to detect changes associated with LD-IL-2 treatment, we decided to maximise the number of T and NK cells from the patients treated with the 3-day interval dosing regimen. We opted for a longitudinal design comparing three different timepoints (Days 0, 27 and 55) from each of these donors treated with $\geq 200,000$ IU/m² dose to decrease experimental variation and thereby increase the power to identify IL-2 induced alterations in a dosing range that had previously been shown to sustain alterations in specific T and NK cell populations (see Seelig E et al. *JCI Insight* 2018). In hindsight, we agree that it would have been beneficial to include intermediate time points between Day 27 and Day 55 to assess the fine dynamics of the anti-inflammatory signature induced by IL-2 treatment but, per trial design, such samples from intermediate time points were not collected from the DILfrequency participants.

We therefore profiled all 11 donors treated with $>300,000$ IU/m² doses and an additional 2 treated with the lower 200,000 IU/m² dose. Of the remaining donors treated with doses $>200,000$ IU/m² every 3 days, we only had five with available PBMC samples (three of which

overlapping with the 13 patients selected from the multiomics study), which were reserved for the follow-up assays described in the revised version of the manuscript. Furthermore, we also included an *in vitro* stimulated condition (using PMA+I) to assess changes in the activation profile and cytokine production. This experimental design generated a large multiomics dataset that maximised the number of cells profiled per donor. We also note that we observed a strong consistency between different donors, which illustrates the power of this dataset to identify compositional and transcriptional changes.

Findings 1. At the level of Treg and NK cells including the finding that low dose IL-2 expands thymic-derived Helios+ Treg and the CD56br NK cell subset, this study only confirms what is already known in T1D and other autoimmune disease treated with LD IL-2, and thus does not add much novel aspects to this (eg von Spee-Mayer Ann Rheum Dis 2016, Humrich Lancet Rheumatol 2019, He Ann Rheum Dis 2019). Apart from this, analysis of Helios expression and of other Treg markers by classical multi-color flow cytometry would certainly have produced more accurate data on frequencies and numbers of such cell subsets than by using single-cell multiomics.

We agree that the analysis of the Treg intracellular markers FOXP3 and HELIOS is critical to stratify the different Treg subsets. Unfortunately, we only had one PBMC aliquot available from each visit, which we prioritised for the multiomics analysis. As referred above in the response to Methodology #1, this decision was also influenced by our previous data, showing a high sensitivity of our targeted multiomics technology to resolve the Treg heterogeneity and especially the distinct subsets expression HELIOS and/or FOXP3.

To validate our multiomics data, using previously collected but unpublished FACS data from our IL-2 dosing studies, we have now assessed the intracellular expression of FOXP3 and HELIOS in the CD127^{low}CD25^{hi} population by flow cytometry in both our single and multiple dosing FACS replication cohorts (see new **Supplementary Fig. 10**). Consistent with our multiomics data, we found a specific reduction in the proportion of memory FOXP3-HELIOS-Teff cells within the CD127^{low}CD25^{hi} population, which was sustained for up to 5 days after dosing in the single-dosing patients (see **Supplementary Fig. 10b**). When normalised to the percentage of total CD4⁺ T cells, these results equated to a very noticeable increase in the frequency of both FOXP3⁺HELIOS⁺ and FOXP3⁺HELIOS⁻ for up to 5 days after treatment, while there was no alteration in the total frequency of FOXP3⁻HELIOS⁻ Teffs (see **Supplementary**

Fig. 10d). Furthermore, in the multiple-dosing cohort, the IL-2-induced reduction FOXP3-HELIOS⁻ cells within CD127^{low}CD25^{hi} T cells, which was sustained during the treatment period (see **Supplementary Fig. 10c**), translated into a selective enrichment of the FOXP3⁺HELIOS⁺ and FOXP3⁺HELIOS⁻ Tregs subsets (see **Supplementary Fig. 10e**), which mirrors the results obtained from the multiomics analysis. We note the strong decrease of all CD127^{low}CD25^{hi} T cell subsets at day 1 after treatment in the single-dose participants, which is consistent with the observed extravasation of CD127^{low}CD25^{hi} T cells observed in the DILT1D study (see Todd JA et al. *PLoS Med* 2016).

Together these data validate the results obtained by the multiomics approach. Moreover, they highlight the increased resolution of this single-cell approach to resolve the functional Treg subsets in a more granular fashion compared to the classical flow cytometric characterisation. In particular, it allows us to dissect the different functional T cell subsets that compose the FOXP3-HELIOS⁻ CD127^{low}CD25^{hi} T cell population and reveals a selective effect of LD-IL-2 treatment on the reduction of specific subsets corresponding to activated CD25⁺ FOXP3⁻HELIOS⁻ Tregs (most notably activated Tfh cells), which could not have been discriminated through the flow-cytometric analysis.

Findings 2. The finding that LD IL-2 is capable to decrease Tfh subsets is also not entirely new, in contrast to the claim of the authors, as this was previously described in SLE patients treated with LD IL-2 therapy (He Nat Med 2016, Humrich Lancet Rheumatol 2019). Given the rather marginal reductions (Fig. 3f) of IL-21 producing CD4⁺ T cell subsets and the high variability, which was found by the single cell multiomics approach, I think it would be necessary to confirm this finding by other approaches, eg is there also a reduction of IL-21 levels in plasma (if measurable) or of IL-21 producing CD4⁺ T cells assessed by flow cytometry in PMA/ionomycin stimulated cells.

Please see response to the Methodology # 3 comment above for further details. As described, we have refrained from claiming novelty of the Tfh modulation finding. We now provide better context of previous data having shown a similar decrease in Tfh cells by FACS in response to LD-IL-2 as well as functionally related marginal zone B cell (He J et al. and Humrich JY et al.) in SLE. We opted to highlight the added value of the single-cell multiomics approach to further dissect Tfh cell differentiation and to show that the observed results are driven by a specific reduction of the more differentiated IL-21-producing Tfh cells.

Although we agree that the differences that we observe in T1D patients in this study are relatively small, we believe that the consistency with previous data (obtained using different methodologies and in different diseases) strongly points to a key role of LD-IL-2 in modulating germinal centre reactions. Unfortunately, due to lack of available patient samples, we are not able to replicate our results by FACS by measuring the intracellular protein expression of IL-21 after *in vitro* stimulation with PMA + ionomycin. This also cannot be achieved via the detection of IL-21 levels in plasma/serum, as the assay does not reach the sensitivity required to detect the levels present in T1D patients. Instead, we have replicated the multiomics results by analysing previously collected flow cytometry data from each visit of the 18 DILfrequency participants as well as newly generated data from the remaining five participants that were used for the intracellular immunophenotyping. Importantly, both methods corroborated our findings using the multiomics approach (please see **Reviewer Fig. 1** and the response to the Methodology # 3 comment above for further details).

Findings 3. The findings on CD8⁺ T cells are of interest and appear also novel. I would discuss these findings a bit more in detail, also in the context of T1D.

We agree with the reviewer that these results within the CD8⁺ T cell population are worth further investigation. Particularly considering the role of the innate-like CD8⁺ MAIT and V_{g9}V_{d2} T cell subsets in anti-viral and anti-bacterial defence (see Provine N et al. *Ann Rev of Immunol* 2020). These data suggest a role of IL-2 immunotherapy in the recruitment of MAIT and V_{g9}V_{d2} cells to tissues following treatment, thereby increasing defence against viral and bacterial infections (see Tao H et al. *Nat Comms* 2021) and may provide a mechanism for the observed decreased incidence of viral infections in SLE patients undergoing LD-IL-2 immunotherapy (see Zhou P et al. *PLoS Pathogens* 2021).

We have added the following text to the Discussion:

“We also observed a decrease in circulating innate-like CD8⁺ MAIT and V_{g9}V_{d2} T cell subsets during treatment and one month after. Since these cells function in anti-viral and anti-bacterial defence⁵⁶, our findings suggest a role of IL-2 immunotherapy in the recruitment of MAIT and V_{g9}V_{d2} cells to tissues following treatment, thereby increasing defence against viral

and bacterial infections⁵⁷. This could provide a mechanism for the observed decreased incidence of viral infections in SLE patients undergoing LD-IL-2 immunotherapy⁵⁸.”

Findings 4. From my point of view, the most interesting and also novel finding of this study was the sustained anti-inflammatory signature present 4 weeks after the treatment including the up-regulation of CISH in Treg and other cells and the downregulation of TNF induced genes. However, as these findings have not been further addressed in detail by other approaches or by correlating them with clinical responses (why are those not shown or mentioned?), they remain only of a descriptive nature. I also miss in the discussion the relation of these findings with T1D pathophysiology.

Overall, I feel that the use of the rather complex single cell multiomics approach in this setting did only provide little new and also rather preliminary information on the mode of action of IL-2, which needs to be confirmed somehow by other approaches.

We agree that the most novel and clinically relevant finding from this study is the identification of a broadly expressed anti-inflammatory signature that can be detected one month after the cessation of treatment. We understand that the key follow-up question is to address the mechanism underpinning this biological response and understand better the specific contribution of IL-2. Correlating the induction of this transcriptional signature with a marker of clinical response would be ideal. The classical marker of disease progression is the rate of decline of C-peptide levels in blood, which require a long sampling interval to properly determine the slope of the decline. In the DILfrequency study there were no clinical outcome measures, and therefore these data were not available for these patients. However, a current trial of LD-IL-2 in children (ITAD; ClinicalTrials.gov Identifier: NCT03782636) has just completed sample collection and will have associated data for C-peptide decline as well as blood sugar levels. The findings from our current multiomics analysis will be critical to design appropriate follow-up experiments in both ITAD and other low-dose IL-2 immunotherapy studies in other diseases, and to focus on the potential longer-term effects of therapy. These later timepoints have been mostly ignored in previous studies, therefore precluding use of currently available data to validate our findings. We have edited the discussion to refer to the ITAD trial and to the need to validate our findings in other studies to better understand the mechanism by which LD-IL-2 immunotherapy elicits a long-term anti-inflammatory cellular/phenotypic environment and clinical responses.

In addition, as discussed in the General Comment #2 point above, we now also provide additional support for the identified transcriptional signature. We replicated the signature in an independent longitudinal cohort of convalescent COVID-19 patients and showed the dynamics of the expression of the signature over time. These data provide important support for the inflammatory/anti-inflammatory role in the context of COVID-19 and LD-IL-2 immunotherapy, respectively as well as for the longevity of the gene expression changes. We acknowledge that these data do not directly demonstrate the role of IL-2 in the mechanism of action of this response in COVID-19 patients but justify its physiological relevance and the need for further investigation. We now explain better the putative mechanism underlying the observed transcriptional signature and as suggested by the reviewer discuss how it could impact on the pathophysiology of T1D.

Findings 5. Dose responses: I think it is not justified to speak about dose dependent effects, considering that there are often only 2 data points (e.g., Fig 6d, Fig. S9e) for the lowest and highest dose in the analyses and that there is a high inter-individual variability in the multiomics data.

We acknowledge that our study design may not be well suited for testing dose-dependent effects, given that nine out of the 13 participants belong to the same dose group. Nevertheless, the correlation between dose levels and Day 55 signature score changes (Fig. 6d) still reached statistical significance ($P = 0.0126$) under a linear regression model. We also applied the Spearman rank-order correlation test, an alternative, more robust statistical test that does not assume a linear relationship, over the same data. This confirmed the positive correlation between the two variables, though with marginal statistical significance ($P = 0.0485$). Therefore, we believe including this correlation in our results will be informative for future investigations about the long-lasting effects of LD-IL-2. Considering the small sample size and high inter-individual variability, we have refrained from suggesting possible dose-dependent mechanisms underlying this observation.

Conclusions/Discussion:

Many of the conclusions in the discussion seem to be speculative, eg that this 3-day interval dosing regimen is better than daily or cyclic regimens. This does not fit to what was found in SLE patients by the use of a repetitive 5-day LD IL-2 regimen which was very effective in selectively expanding Treg without any hints for desensitization (Humrich Lancet Rheumatol 2019). To address this issue a direct comparison between the regimens would be required. In particular, the COVID-19 data interpretation is highly speculative and should be removed as suggested above. From my personal view I would not suggest to give IL-2 to patients with acute and severe hyperinflammation as observed in COVID-19 infections, as this could be like putting oil into fire.

The question of dosing regimen for low-dose IL-2 immunotherapy is an interesting topic that requires further studies. We agree with the reviewer that the claims of improved therapeutic efficacy of the interval dosing regimen were overstated in the original version of the manuscript. We have now revised the manuscript (and the Discussion section in particular) accordingly to refrain from claiming such clinical benefits. Instead, we refer to the need to better understand the differences between dosing regimens, as these will likely be key for their disease-specific application and to the demographics and treatment duration of the target patient population. This is likely to be different in the setting of treating patients with active disease or in prevention therapy and will need to be properly addressed in comparative studies.

With regards the interpretation of the COVID-19 data, we agree with the reviewer. We certainly did not mean to suggest treating COVID-19 patients with low-dose IL-2 during the acute phase of the disease. As we have previously discussed in the context of maximising the therapeutic benefit of LD-IL-2 immunotherapy, we believe that LD-IL-2 is ideally suited to restore and maintain a regulatory environment, especially in relapsing/remitting diseases, to increase the time between flares (and reduce its severity). We completely agree that during periods of hyperinflammation, increased IL-2 availability is not only redundant, but may lead to further activation pathogenic cells and increased production of pro-inflammatory cytokines (i.e. "putting oil into fire").

We have edited the discussion to make this point clearer and to specifically mention that a role for LD-IL-2 immunotherapy should only be considered in convalescent patients in the context of prevention of long-COVID. This will obviously need to be further investigated in the appropriate experimental design and will require a much better understanding of the

mechanism causing the long-term sequelae of COVID-19 and which patients are likely to suffer from them.

Terminology 1. The terminus “effector” T cell is a bit misleading in this context, as those non-Treg CD4+ T cells also contain many naïve T cells. I would suggest using the term conventional T cell (Tcon or Tconv) instead.

We agree that the use of ‘effector T cell’ designation is not clear in the manuscript, as it encompasses all naïve T cell subsets. As suggested by the reviewer, we have replaced it throughout (manuscript and figures) by the designation of conventional T cell (Tconv), which is a more objective designation of the cells not corresponding to the Treg population. We reserve the use of the effector T cell (Teff) designation for memory subsets with proper effector function.

Terminology 2. I would not use the term “depletion” for describing the reduction of cell subsets, as this suggests that these cells are actively removed or killed, but most likely there is just less generation or differentiation of these subsets.

We thank the reviewer for the suggestion. We have now edited the text and replaced the term “depletion” by “reduction’ to avoid erroneous interpretations.

Reviewer #3:

1. [...] In the UMAPs depicting the clustering of unstimulated and stimulated cells (Supplementary Figure 2) is there any population/cluster appearing after treatment (or any time/dose-specific population?)

Apart from significant compositional changes in several subsets as we described in the manuscript, we did not observe any cluster that was exclusively to a specific time point or dose. We believe this likely reflected the effective integration of samples from different participants and time points, considering that such sample-specific clusters often result from batch effects in practice.

2. Both in the case of Tfh and Treg have the authors tried to perform a trajectory analysis confirming the differentiation of the populations?

We thank the reviewer for this suggestion. We have now performed a trajectory analysis using Slingshot (Street K et al. BMC Genomics, 2019), which enabled the quantitative identification of Treg and T_{FH} differentiation trajectories (**Supplementary Fig. 8**). These results confirmed the compositional changes that we described, namely the increase of naïve Treg cells and the reduction of IL-21-producing T_{FH} cells. Further details about these results can be found in the corresponding results and methods sections.

3. Why the authors did not decide to regress out cell cycle genes from the analysis instead of excluding the clusters of cycling cells in a later stage of the analysis?

Although we excluded cycling cell clusters from the sub-clustering of Treg and Tconv cells, we actually included cycling cells in all downstream analysis. This enabled us to identify a significant decrease of cycling Tregs on Day 27 (**Supplementary Fig. 3b**) and, correspondingly, down-regulation of cell cycle genes in Tregs and CD56^{br} NK cells.

We made the decision to exclude cycling cell clusters from sub-clustering based on the following observations: (i) cycling cells were relatively rare in numbers; (ii) cycling cells clusters had well-defined mRNA markers; (iii) the phenotypical differences between cycling

cells and their non-cycling counterparts were potentially larger than the heterogeneity within each major cell populations. We have now updated the Methods section to describe these considerations more clearly.

Meanwhile, we believe that regressing out the cell cycle genes at an early stage is an equally valid alternative compared to our approach, and should have minimal impact on our primary findings, considering the relatively small number of cycling cells (0.72% of total unstimulated cells). However, we believe regressing out the cell cycle genes would not allow us to draw conclusions on cycling cells as described above

4. Supplementary figure 2 legend sentence "Untagged cells in b (labelled as "Tag N/A" correspond to ..." should be "Untagged cells in a" ...)

We thank the reviewer for pointing out this inconsistency. We have now fixed this as suggested.

5. In the printed version of the paper (A4) Labels in Supplementary figure 5 (boxplots) are not readable

We have now updated Supplementary Fig. 5 (now **Supplementary Fig. 6**) to ensure the visibility of the labels. We have separated the data in that figure into two supplementary Figures to provide more space and improve legibility of the panels.

Reviewer Figure 1. Immunophenotyping the CD4⁺ T follicular helper (T_{FH}) subset by FACS.

(a) Gating strategy for the delineation of CD4⁺ T_{FH} cells by FACS. (b,c) Variation (depicted as the % change from pre-treatment baseline levels) in the relative frequency of CD4⁺ T_{FH} cells. Frequency of CD4⁺ T_{FH} cells was determined by flow cytometry in whole-blood from the 18 DILfrequency participants treated with the 3-day interval dosing schedule (b) or in a subset of 5 DILfrequency donors selected for intracellular immunophenotyping (IP) - treated with 200,000-320,000 IU/m² dose range every three days. Data shown depict the average (± SEM) variation of the assessed immune subsets at each visit.

REVIEWERS' COMMENTS

Reviewer #1 (Remarks to the Author):

While the authors provide much more evidence to support the conclusion, the data and logic in the current manuscript are still flawed. As single-cell targeted RNA-seq based on the BD platform enables single-cell whole transcriptome sequencing based on the same sample, transcriptome-wide data should be added to dissect the effects of IL2 treatment. Although a new dataset of COVID-19 was added, the logical linkage between IL2 treatment and COVID-19 is still absent.

Reviewer #2 (Remarks to the Author):

I am quite happy with the authors' responses and their detailed explanations, and with the changes and additional data in the revised version of the manuscript.

I have only some minor suggestions:

References:

1. Ref 29 could be cited already in the 1st sentence of the introduction (line 49), as this also belongs to the clinical trials in autoimmune diseases mentioned here. A very recent phase 2 RCT in SLE could also be included here (Humrich et al. Ann Rheum Dis 2022, PMID 35973803).
2. The work by Hirakawa et al. (JCI Insight 2016, PMID 27812545) addressing the effects of LD IL-2 on Helios+ Treg and NK cells in GvHD could be cited in the respective paragraphs.

Figure

For reasons of clarity, I suggest to include the Reviewer Figure 1 showing the Tfh gating in the supplemental appendix.

Method section

As shown data were derived from participants of a clinical trial, I think it is necessary to add a sentence on ethics (informed consent/Helsinki/ICH-GCP) and to provide the trial registration number in the section "Study design and participants".

Reviewer #3 (Remarks to the Author):

Authors addressed all my concerns and now the manuscript has significantly improved.

NCOMMS-22-13925A

Point-by-point replies to Reviewers' comments

Reviewer #1

1. While the authors provide much more evidence to support the conclusion, the data and logic in the current manuscript are still flawed. As single-cell targeted RNA-seq based on the BD platform enables single-cell whole transcriptome sequencing based on the same sample, transcriptome-wide data should be added to dissect the effects of IL2 treatment. Although a new dataset of COVID-19 was added, the logical linkage between IL2 treatment and COVID-19 is still absent.

We have now edited the manuscript to remove all data related with the COVID-19 analyses. We agree that this provides more focus on the mechanism of LD-IL-2 treatment and the potential clinical benefits. It also mitigates the main limitation of the manuscript, which was the absence of a direct mechanism linking LD-IL-2 treatment and the induction of the observed gene expression changes in COVID-19. This mechanism is likely complex and not solely regulated by IL-2, which we agree distracts from the main topic of the manuscript.

With regards whole transcriptome sequencing, while the Reviewer is correct that it is possible to generate using the BD Rhapsody system, it utilises a different protocol after total poly-A mRNA capture on the cell-capture beads. In our targeted assay we have amplified only the genes present on the panel, and therefore only have cDNA available from those specific regions. To generate whole transcriptome data we would need to go back to new cell aliquots to amplify total cDNA using a random priming method, which we do not have available for the same patients and visits.

Reviewer #2:

References:

1. Ref 29 could be cited already in the 1st sentence of the introduction (line 49), as this also belongs to the clinical trials in autoimmune diseases mentioned here. A very recent phase 2 RCT in SLE could also be included here (Humrich et al. Ann Rheum Dis 2022, PMID 35973803).

We thank the Reviewer for the suggestion and have revised the first paragraph of the introduction to reference these clinical studies.

2. The work by Hirakawa et al. (JCI Insight 2016, PMID 27812545) addressing the effects of LD IL-2 on Helios+ Treg and NK cells in GvHD could be cited in the respective paragraphs.

We have edited the discussion section to include a reference to this work, as it provides additional validation of the specificity of LD-IL-2 on the expansion on HELIOS+ Tregs and CD56br NK cells, even in a different disease and using a daily dosing regimen.

Figure

For reasons of clarity, I suggest to include the Reviewer Figure 1 showing the Tfh gating in the supplemental appendix.

For completeness, we have now added these FACS data to the results section. This figure is now shown as Supplementary Figure 12 in the manuscript to provide a more comparable FACS analysis of the circulating Tfh cells in T1D to the one reported in SLE.

Method section

As shown data were derived from participants of a clinical trial, I think it is necessary to add a sentence on ethics (informed consent/Helsinki/ICH-GCP) and to provide the trial registration number in the section "Study design and participants".

We have added a section at the beginning of the Methods to include a statement relating to the study approval, ethics and informed consent information. We now also provide the trial registration ID and study protocol for both DILT1D (single dosing) and DILfrequency (multiple dosing) studies.